# A universal reading network and its modulation by writing system and reading ability in French and Chinese children

Xiaoxia Feng[1,2], Irene Altarelli[1,3], Karla Monzalvo[1], Guosheng Ding[2], Franck Ramus[4], Hua Shu[2], Stanislas Dehaene[1,5], Xiangzhi Meng[6,7]*, Ghislaine Dehaene-Lambertz[1]*

[1]Cognitive Neuroimaging Unit, CEA DRF/I2BM, INSERM, NeuroSpin Center, Université Paris-Saclay, Gif-sur-Yvette, France; [2]State Key Laboratory of Cognitive Neuroscience and Learning & IDG/McGovern Institute for Brain Research, Beijing Normal University, Beijing, China; [3]Université de Paris, LaPsyDÉ, CNRS, Paris, France; [4]Laboratoire de Sciences Cognitives et Psycholinguistique (ENS, CNRS, EHESS), Ecole Normale Supérieure, PSL Research University, Paris, France; [5]Collège de France, Université PSL Paris Sciences Lettres, Paris, France; [6]School of Psychological and Cognitive Sciences, Beijing Key Laboratory of Behavior and Mental Health, Peking University, Beijing, China; [7]PekingU-PolyU Center for Child Development and Learning, Peking University, Beijing, China

**Abstract** Are the brain mechanisms of reading acquisition similar across writing systems? And do similar brain anomalies underlie reading difficulties in alphabetic and ideographic reading systems? In a cross-cultural paradigm, we measured the fMRI responses to words, faces, and houses in 96 Chinese and French 10-year-old children, half of whom were struggling with reading. We observed a reading circuit which was strikingly similar across languages and consisting of the left fusiform gyrus, superior temporal gyrus/sulcus, precentral and middle frontal gyri. Activations in some of these areas were modulated either by language or by reading ability, but without interaction between those factors. In various regions previously associated with dyslexia, reading difficulty affected activation similarly in Chinese and French readers, including the middle frontal gyrus, a region previously described as specifically altered in Chinese. Our analyses reveal a large degree of cross-cultural invariance in the neural correlates of reading acquisition and reading impairment.

*For correspondence:
mengxzh@pku.edu.cn (XM);
ghislaine.dehaene@cea.fr (GD-L)

**Competing interests:** The authors declare that no competing interests exist.

## Introduction

A large proportion of published studies on reading concern English, even though this language may be considered as an outlier within alphabetic writing systems, compared to Finnish or Italian for example, due to its highly opaque grapheme-phoneme correspondences. Thus, international efforts should progressively extend the results obtained in English subjects, or more generally in Western languages, to other writing systems and languages, notably during childhood. *Daniels and Share, 2018* listed 10 dimensions that might affect reading acquisition and dyslexia phenotypes, and that fall into three main classes: the structure of the oral language, the complexity of the visual shapes, and the translation rules between those two domains. With regard to language, the number of phonemes, the syllabic structure and the complexity and regularity of morphological markers can modulate the ease with which children construct explicit representations of speech, which need to be

converted in, or deduced from, writing (*Goswami, 2008*). The visual shapes of letters and characters also vary in number, uniformity, and complexity (*Daniels and Share, 2018*). Finally, the grapheme-to-phoneme correspondences vary on several dimensions across writing systems, including granularity, complexity, transparency, and consistency (*Daniels and Share, 2018*). All these factors may influence the speed and effectiveness with which children learn to read, and at least one of these dimensions, orthographic transparency, has been robustly reported to affect reading acquisition in Western languages (*Seymour et al., 2003*), with a reported impact on brain activation in the adult reading circuit (*Paulesu et al., 2000*).

Beyond these surface differences, the fundamental logic of reading remains the same from one writing system to another: all of them comprise a restricted number of visual symbols whose combinations allow to access the spoken language network through vision. Although accessing linguistic information from the visual system is a natural possibility for the brain, as demonstrated by the capacities for image naming, lip reading or sign language, writing introduces an additional step that involves converting an arbitrary visual form into speech. Grapheme-phoneme correspondences are arbitrary and have been shown to depend on a precise region of the visual system, the visual word-form area (VWFA) and on the posterior superior temporal cortex, as shown by numerous brain imaging studies in adult readers (*Baker et al., 2007*; *Dehaene et al., 2015*; *Stevens et al., 2017*). However, most of those brain imaging studies suffer from the bias noted above: they concern mainly Western alphabetic languages and mainly adults. Despite the existence of a few previous investigations (*Chee et al., 1999*; *Nakamura et al., 2012*; *Rueckl et al., 2015*; *Szwed et al., 2014*; *Xu et al., 2017*), controversy still surrounds the question of whether different cognitive and neural processes are involved in reading non-alphabetic material, or whether reading is based on a similar network regardless of the target language, with only minor variations in the degree of involvement of the different nodes of the network according to the graphic complexity and consistency (*Paulesu et al., 2000*) or to the linguistic grain size that predominates in a given writing system (*Perfetti, 2003*; *Ziegler and Goswami, 2005*). For instance, Chinese characters, like letters, eventually map onto phonology, but they do so at the syllable level and with a considerable degree of irregularity, with (in most cases) no parts in a character corresponding to phonological segments such as phonemes.

An additional source of bias in the literature is that most publications on brain imaging have focused on adults, that is, reading experts. It is entirely conceivable that scanning children during the acquisition of reading would yield different results. The transient mechanisms of learning, in an immature and inexperienced brain, could be based on different mechanisms than those observed in the mature brain (*Kersey et al., 2019*). This is especially the case for a highly cultural, education-dependent activity such as reading (*Dehaene and Cohen, 2007*). Children may rely on different and possibly broader regions of the brain before converging onto the adult expert network for reading, for instance transiently recruiting parietal regions for effortful reading (*Dehaene-Lambertz et al., 2018*; *Martin et al., 2015*). Conversely, they may also enlarge their responses with skill acquisition, as described in the fusiform region (*Olulade et al., 2013*). Furthermore, given the differences between writing systems outlined above, initial reading might rest upon different brain areas before converging to a common circuit. Therefore, functional neuroimaging studies in young children, comparing reading acquisition in different writing systems, although difficult, are highly desirable.

Another approach to studying the universality and specificity of the neurocognitive bases of reading is to investigate whether reading-impaired children from alphabetic and non-alphabetic languages exhibit similar brain anomalies. Neuroimaging studies of dyslexia have revealed common neural deficits in different alphabetic languages, with consistently decreased activation (*Martin et al., 2016*; *Richlan et al., 2009*) and reduced gray-matter volumes (*Linkersdörfer et al., 2012*; *Richlan et al., 2013*) in several left-hemispheric posterior regions, including the left temporo-parietal and left ventral occipitotemporal regions. According to two recent meta-analyses, the left ventral posterior occipitotemporal cortex (including the Visual Word-Form Area, VWFA) appears to be the most reproducible and consistent site exhibiting hypoactivation in dyslexic individuals across several alphabetic writing systems regardless of orthographic depth (*Martin et al., 2016*). Given its sensitivity to visual features (e.g. line junctions) and its efficient reciprocal projections to language areas (*Bouhali et al., 2014*; *Hannagan et al., 2015*; *Saygin et al., 2016*), this area has been proposed as one of the candidates for a universal effect of reading disability (*Martin et al., 2016*; *Pugh, 2006*).

By contrast, several neuroimaging studies of Chinese dyslexic children have emphasized the differences between Chinese and alphabetic languages, underscoring the role of the left middle frontal gyrus (LMFG) (*Liu et al., 2013*; *Siok et al., 2004*; *Siok et al., 2009*): Chinese dyslexic children showed a decreased activation in the LMFG compared with typical readers (with eight 11-year-old children in each group) during a homophone judgment task (*Siok et al., 2004*). This decreased activation in the LMFG was replicated by *Siok et al., 2008*, with twelve 11-year-old children in each group. It was associated with smaller gray-matter volume at the same location, and no other functional or structural differences was found in the regions singled out by other studies in alphabetic languages. These results were interpreted as showing a clear dissociation of the biological basis of reading disabilities between alphabetic and logographic writing systems. However, these findings contradict behavioral data that show similar profiles in Chinese and alphabetic-language dyslexics (*Goswami et al., 2011*; *Ziegler and Goswami, 2005*) and similar predictors of reading abilities in both writing systems. For instance, phonological awareness and morphological awareness (lexical compounding) in 4-year-old Chinese children predict character recognition at 11 years of age, while naming speed (RAN) and vocabulary predict reading fluency (*Su et al., 2017*). These results are consistent with those obtained in alphabetic languages, which link phonological awareness to reading accuracy and RAN to reading fluency (*Landerl et al., 2013*; *Moll et al., 2014*). Moreover, another fMRI study revealed remarkably few differences in brain activity between 11 English and 11 Chinese dyslexic adolescents (13 to 16 years) once all confounding variables (e.g. stimuli and task in fMRI) were controlled for (*Hu et al., 2010*). It is worth noting that previous fMRI studies of Chinese children with reading disability used a variety of tasks, e.g. picture and semantic matching (*Hu et al., 2010*), homophone judgment (*Siok et al., 2004*), font-size perceptual matching (*Siok et al., 2009*) and a morphological task (*Liu et al., 2013*), which may partially explain the inconsistency of the findings between studies.

Therefore, to clarify the question of the commonalities and specificities of reading acquisition across different writing systems, we used a similar experimental protocol in 96 Chinese and French 10-year-old readers (48 in each language), with different reading proficiency. Half of the children had reading difficulties, meeting the criteria for dyslexia in French children (more than 2 years of reading delay) though not as marked in Chinese children. (*Table 1*). As in our previously published fMRI studies of reading (*Dehaene-Lambertz et al., 2018*; *Monzalvo and Dehaene-Lambertz, 2013*), all children performed the same passive viewing task with words, faces, and houses, with the mere goal to detect an occasional target star. We studied the effect of reading proficiency in whole-brain analyses and in the specific ROIs highlighted in the literature, in both Chinese and French children. The two analyses are complementary: whole-brain analyses can reveal any regions with differences between Chinese and French children, or between children with and without reading difficulties, including at unexpected brain sites, while ROI analyses have a better sensitivity to detect small differences between groups by summing voxel activity in the cluster and decreasing the number of repeated measures and thus the severity of the correction for multiple comparisons.

Because group analyses leave open the possibility that the observed group differences might be due to spatially more variable activations in poor readers relative to typical readers, we also performed individual-based analyses and compared the location and activation values of the most responding voxels in each child. Finally, because classical analyses may mask the presence of fine-

**Table 1.** Characteristics of the four groups.

| | Chinese | | French | |
|---|---|---|---|---|
| | Typical readers | Poor readers | Typical | Poor readers |
| Sample size | 24 | 24 | 24 | 24 |
| Age in months (SD) | 123 (11) | 123 (10) | 123 (11) | 123 (10) |
| Sex | 13M/11F | 16M/8F | 13M/11F | 16M/8F |
| Reading ability (CI 95%) | 0.67 (0.49 ~ 0.86) | −1.74 (−1.95 ~ −1.54) | 0.73 (0.39 ~ 1.07) | −2.16 (−2.38 ~ −1.94) |

The online version of this article includes the following source data for Table 1:

**Source data 1.** Demographic information of the four groups.

grained activity patterns that are specific to a given subject or a given category, we also quantified the stability of subject-specific activation patterns within reading-related ROIs using multivariate pattern analyses. The goal of these analyses is to circumvent the blurring effect of group analyses, which may hinder the discovery of genuine but more dispersed activations in poor readers relative to typical readers.

In addition to activations to words, we were also interested in how reading acquisition, in French and Chinese, may differently reorganize the ventral visual areas. Fusiform regions have a distinct maturation profile than more medial regions (*Gomez et al., 2017*) and face-specific activations expand slowly with age (*Golarai et al., 2007*; *Golarai et al., 2015*). Several studies in alphabetic languages suggest that, during reading acquisition in children, words, and faces may compete for cortical territory within the left fusiform gyrus (*Centanni et al., 2018*; *Dehaene-Lambertz et al., 2018*; *Hervais-Adelman et al., 2019*; *Li et al., 2013*; *Ventura et al., 2013*). Given the complexity of Chinese characters and their frequently reported bilateral activation, we investigated how face activation might be differently modulated by different reading abilities in Chinese and French children. We thus performed the same analyses for faces than those described above for words and also considered more precisely the development of the anterior-posterior gradient of activations for these two categories.

## Results

### Behavioral Results

Within the scanner, Chinese children responded faster to the target star than French children (main effect of Language: $F_{(1,92)}=60.94$, $p<0.001$). There was no significant effect of Reading ability ($F_{(1,92)}<1$) nor Language $\times$ Reading ability interaction ($F_{(1,92)}<1$) (Chinese typical readers: $534.95 \pm 71.47$ ms, Chinese poor readers: $536.13 \pm 72.84$ ms; French typical reader: $661.15 \pm 102.29$ ms, French poor readers: $689.32 \pm 99.35$ ms).

### Whole brain analyses

#### Category-specific activations

We first examined the brain activations to each category (i.e. Words, Faces, and Houses) relative to the other two categories among all participants (see *Figure 1A* and *Table 2*). The [Words > Faces + Houses] analysis yielded the usual reading-related regions: fusiform gyrus, posterior superior temporal region, *planum temporale*, intra-parietal sulcus and inferior frontal regions in the left hemisphere and the posterior superior temporal gyrus in the right hemisphere. We also observed the classic mosaic of category-specific ventral visual areas, with category-specific activation to Houses occupying a medial parahippocampal location, Faces an intermediate fusiform location, and Words a lateral location in the left occipito-temporal sulcus (VWFA). Amygdala responses to Faces were also clearly seen. Those results were seen in each of the four groups of subjects, with the interesting exception that the left VWFA was not seen in both Chinese and French poor readers, contrary to the typical readers, at this classical threshold (voxel-level $p<0.001$, cluster-level uncorrected) (*Figure 1B* and *Supplementary file 3*).

### Reading-related differences

Reading scores across all 96 children were significantly correlated with fMRI activation in the [words > fixation] contrast, in the classical regions of the reading circuit in the left hemisphere (fusiform, superior temporal sulcus, middle frontal region and precentral) plus some of their right counterparts (*Figure 2* and *Table 3*). When the French and Chinese groups were considered separately, similar regions (bilateral FFG, bilateral PCG, bilateral MFG, left STS,) were observed in each of the two languages (*Figure 2B and C*).

We also observed a significant positive correlation, across all children, between reading scores and the face-evoked activation (vs fixation) in the left fusiform gyrus (left [−33 −60 −15], 74 voxels, $Z = 4.31$, $p_{FWE\_corr} = 0.001$). This region was close to coordinates of the classic left FFA (*Scherf et al., 2007*). A small cluster was also observed close to the classic right FFA, but did not survive the cluster-level correction ([45 −48 −15], 12 voxels, $Z = 3.70$).

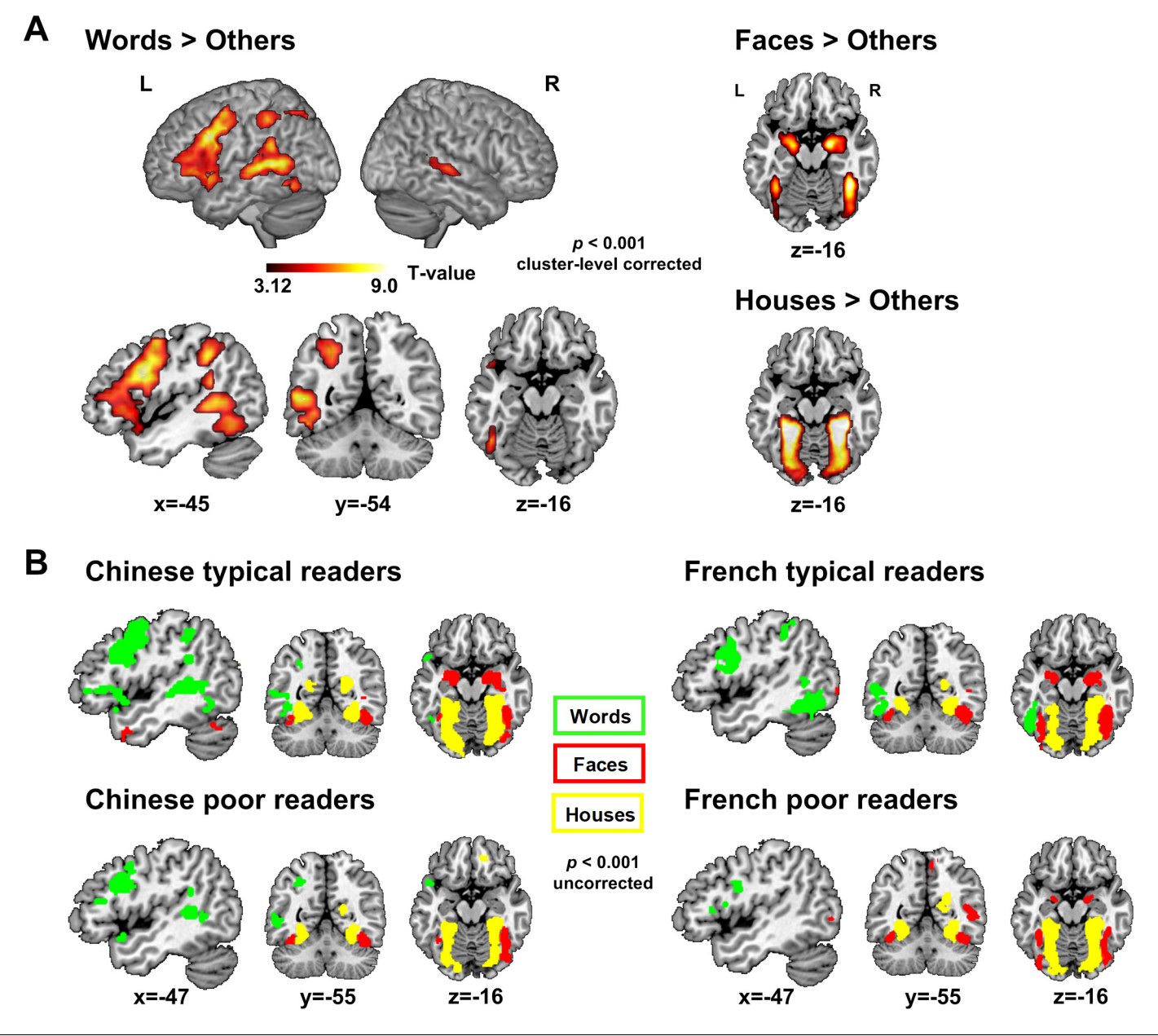

**Figure 1.** Category-specific circuits. (A) Category-specific circuits across all participants (voxel-level p<0.001, cluster-level FWE corrected p<0.05). On the left, the reading circuit identified by the contrast Words > [Faces, Houses] and on the right: Face-selective (Faces > [Words, Houses]) and House selective regions (Houses > [Words, Faces]). (B) Category-specific circuits in each of the four groups (voxel-level p<0.001, cluster-level uncorrected). On the left, category-specific activations in Chinese typical readers (above) and Chinese poor readers (below). On the right, category-specific activations in French typical readers (above) and French poor readers (below). Green: regions selectively activated by words (Words > [Face, House]); Red: regions selectively by faces (Face > [Word, House]); Yellow: regions selectively activated by Houses (House > [Word, Face]).

The online version of this article includes the following figure supplement(s) for figure 1:

**Figure supplement 1.** Language × reading ability ANOVA analysis on children's activation to words.

## Group comparisons

No significant difference was found between typical and poor readers in either direction when analyzing either the [Words > fixation] or the [Words > Faces+Houses] contrast across the whole brain. However, a few voxels reached the voxel-wise statistical threshold (p=0.001) in the left fusiform gyrus, left precentral and left superior temporal sulcus with greater activation in typical readers

**Table 2.** Regions of significant activations for each visual category vs. the two others across all participants.

| Region | MNI coordinates | Peak *p*-value | Peak z-value |
|---|---|---|---|
| Words > others | | | |
| Left inferior frontal gyrus | −48 12 30 | 2.06e-19 | 8.93 |
| Left precentral | −39 0 36 | 1.45e-18 | 8.72 |
| | −51 6 39 | 2.91e-14 | 7.51 |
| Left superior temporal gyrus/sulcus | −57 −30 3 | 2.39e-19 | 8.92 |
| Left middle temporal gyrus | −51 −42 6 | 7.89e-18 | 8.52 |
| Left fusiform gyrus | −48 −57 −15 | 1.69e-17 | 8.43 |
| Left Inferior parietal sulcus | −45 −39 42 | 4.29e-14 | 7.46 |
| Right superior temporal sulcus | 57 −27 3 | 8.94e-10 | 6.02 |
| Faces > others | | | |
| Left fusiform gyrus | −39 −48 −21 | 3.28e-17 | 8.35 |
| Right fusiform gyrus | 42 −54 −18 | 6.14e-26 | 10.47 |
| Right amygdala/hippocampus | 18 −9 −18 | 6.11e-22 | 9.56 |
| Left amygdala/hippocampus | −18 −9 −18 | 3.91e-15 | 7.77 |
| Houses > others | | | |
| Left fusiform gyrus | −30 −48 −6 | 9.01e-53 | 15.24 |
| Right fusiform gyrus | 30 −45 −9 | 2.90e-50 | 14.86 |
| | 27 −63 −9 | 2.40e-22 | 9.65 |
| Left calcarine | −18 −54 9 | 8.66e-10 | 6.02 |

relative to poor readers (see *Figure 1—figure supplement 1A*).To improve the sensitivity to differences between groups and decrease the risk of false negatives, we restricted our analyses to reading sensitive regions defined by the mask comprising all voxels showing a preference for words relative to the two other categories across all participants. Typical readers showed larger activations relative to poor readers in a left precentral cluster (79 voxels, $p_{FWE\_corr}$ = 0.027, Z = 3.69 at [−51 15 33]) and Chinese relative to French in the left intra-parietal sulcus (55 voxels, $p_{FWE\_corr}$ = 0.004, Z = 4.29 at [−30–60 39]) in the words vs. fixation contrast (*Figure 1—figure supplement 1*). No region showed more activation in French children relative to Chinese nor a significant language × reading ability interaction, even when a very lenient voxel-wise threshold of p<0.05 was considered.

To summarize the results so far, our analyses recovered, in a large group of 10-year-old children, the classical activations for words, faces, and houses described in adults. Reading proficiency modulated the response to words in the classical reading circuit and contralateral regions, but also to faces in the fusiform region, in both languages. However, a binary classification of the participants in typical and poor readers was less powerful, recovering a few voxels with significant hypo-activations at expected locations in the poor readers relative to the typical readers, which did not survive corrections for repeated measures. To circumvent this reduced power, we next focused on brain regions which have been reproducibly shown to be under-activated in dyslexics or modulated by Chinese writing, and studied whether and how reading ability and writing system affected their response to words.

## Literature-driven analyses

*Figure 3A* presents all foci reported in four published meta-analyses of dyslexia in alphabetic languages (*Linkersdörfer et al., 2012*; *Maisog et al., 2008*; *Richlan et al., 2009*; *Richlan et al., 2011*) and in four meta-analysis of Chinese typical reading (*Bolger et al., 2005*; *Tan et al., 2005a*; *Wu et al., 2012*; *Zhu et al., 2014*). As seen in *Figure 3A*, dyslexia in alphabetic languages is consistently characterized by dysfunctions in the left occipito-temporal, temporoparietal and frontal regions. All ROIs (except two ROIs in the right hemisphere) fell within the reading circuit identified in

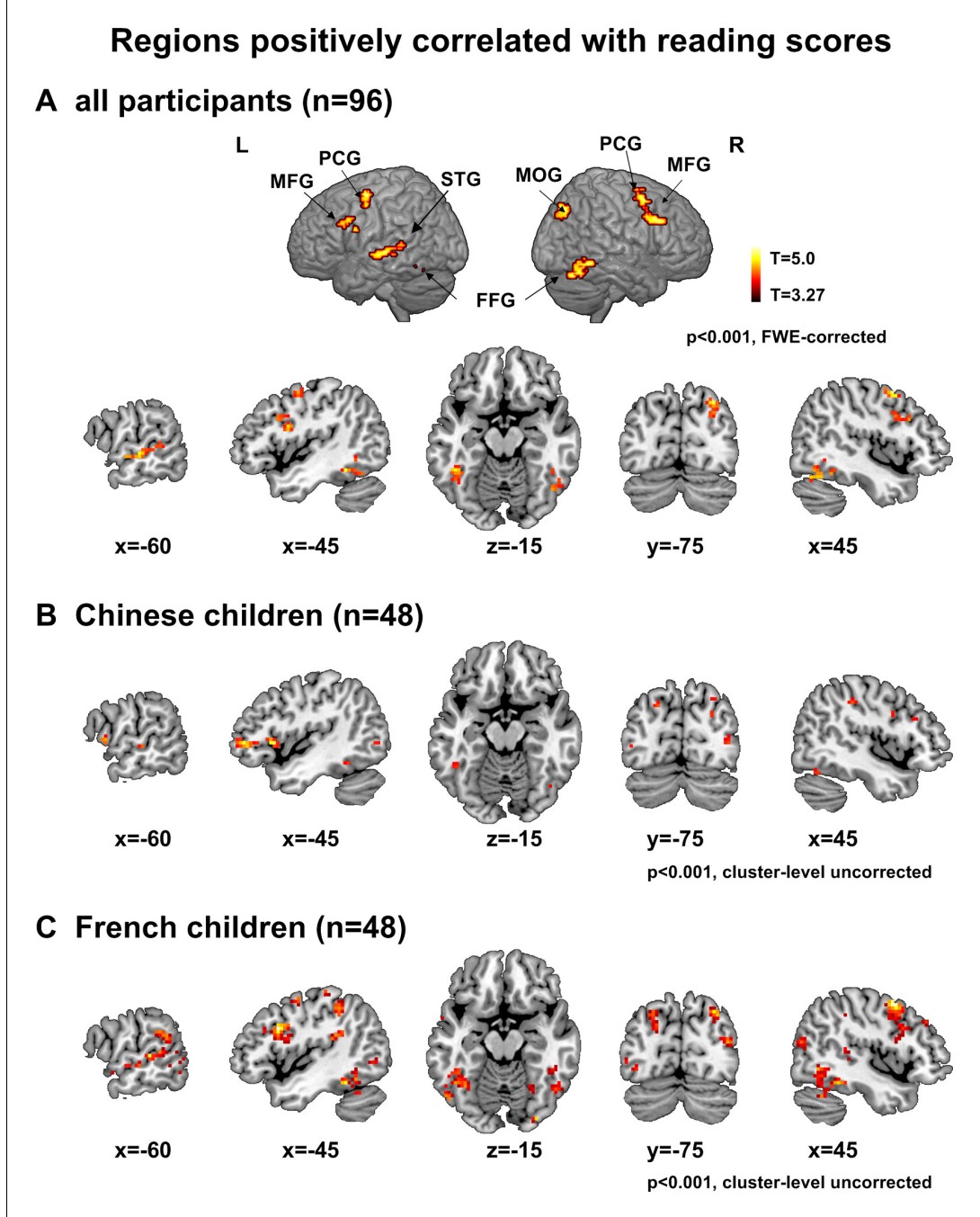

**Figure 2.** Neural correlates of inter-individual variability in reading scores. (A) The figure shows the regions whose activation in the words versus fixation contrast was positively correlated with reading scores across all participants at the whole-brain level (voxel-wise p<0.001 and cluster-wise p<0.05 FWE corrected).(B, C) Regions positively correlated with reading scores in Chinese and French children (voxel-wise p<0.001, cluster-level uncorrected).

our participants (Words > other categories; see *Figure 4* and *Figure 4—figure supplement 1*). *Figure 3B* presents the correlation (FDR-corrected) between reading score and activation to words in these ROIS. This correlation reached FDR-corrected significance in PCG, STS, MTG, and FFG, as well as the MFG and RIOG previously reported in Chinese readers (*Figure 3B*).

We submitted the activation to words relative to fixation in each ROI to a 2 × 2 ANOVA with Language and Reading ability as between-subject factors. Below, we report only the *p*-values that survived an FDR correction over the 13 ROIs.

**Table 3.** Regions significantly correlated with reading scores across all participants at the whole-brain level.

| Region | MNI coordinates | Peak $p$-value | Peak z-value | R-value |
|---|---|---|---|---|
| Left fusiform gyrus | −42 −45 −18 | 6.65e-6 | 4.36 | 0.490 |
| Right fusiform gyrus | 42 −66 −24 | 9.96e-6 | 4.27 | 0.512 |
| Left precentral | −36 −3 57 | 2.75e-6 | 4.54 | 0.467 |
| Right precentral | 54 18 33 | 1.29e-5 | 4.21 | 0.432 |
| Left middle frontal gyrus | −36 12 27 | 1.92e-5 | 4.12 | 0.468 |
| Right middle frontal gyrus | 45 6 54 | 7.72e-6 | 4.32 | 0.460 |
| Left superior temporal sulcus | −57 −24 0 | 1.00e-5 | 4.26 | 0.510 |
| Right middle occipital gyrus | 27 −69 42 | 6.34e-6 | 4.37 | 0.448 |

A significant main effect of Language was observed in the left middle frontal gyrus (F (1,92) =15.23, p<0.001, $p_{FDR\_corr}$ < 0.001), superior parietal lobule (SPL, F (1,92)=8.13, p=0.005, $p_{FDR\_corr}$ = 0.022) and posterior superior temporal gyrus (pSTG, F (1,92)=9.04, p=0.003, $p_{FDR\_corr}$ = 0.020), always due to larger activation in Chinese readers than in French readers (see *Figure 4*).

Significant reduced activation was observed in poor readers relative to typical readers in the left fusiform gyrus (F(1,92) = 21.08, p<0.001, $p_{FDR\_corr}$ < 0.001), middle frontal gyrus (F (1,92)=10.17, p=0.002, $p_{FDR\_corr}$ = 0.009), superior temporal sulcus (F (1,92)=11.88, p=0.001, $p_{FDR\_corr}$ = 0.006), and precentral gyrus (F (1,92)=7.78, p=0.006, $p_{FDR\_corr}$ = 0.020). Importantly, it was the case within each language group, except for the pSTG where a significant difference between typical and poor readers was only observed in French but not Chinese children (*Figure 4*). No ROI showed a significant language × reading ability interaction (see *Figure 4* and *Figure 4—figure supplement 1*), even the pSTG (F (1,92)=2.815, p=0.097 before FDR correction).

We used Bayesian ANOVAs on the Word activation to assess the likelihood of the null hypothesis H0 over H1 in the case of the interaction language × reading ability. As also explained in the methods, the Bayes factor (BF10) is the ratio of the amount of evidence for H1 above H0 (BF01 for H0 above H1). Evidence for H1 against H0 is generally considered as moderate for BF10 ≥ 3, and strong for BF10 ≥ 10. Bayesian analyses provided similar conclusions to frequentist analyses for the main effect of Language and Reading ability (*Table 4*). The BF10 for an effect of language was 80.21 in the MFG, 7.05 in the SPL and 9.91 in the pSTG. The likelihood of an effect of reading ability in the FFG was BF10 = 1528.97 higher than that of the null hypothesis; the same BF10 was 37.51 in the STS, 9.95 in the MFG and 5.63 in the PCG respectively. It was also the case in the post-hoc analyses within each language (*Table 4*). Notably, there was strong evidence of a reading ability effect in the fusiform gyrus (FFG) and moderate evidence in the STS and PCG in both languages. In the pSTG and also paradoxically in the MFG described as a dyslexic marker in Chinese, the evidence of a reading ability effect was strong in French (BF10 = 23.74 and 29.35 for each site respectively) but absent or weak in Chinese children (BF10 < 1 and 1.62). Sensitivity analysis revealed that the Bayes factor stayed about the same for a wide range of prior specifications (Cauchy prior width: 0–1.5) in the FFG, STS, PCG, and MFG for the comparisons between typical and poor readers in each language except for the pSTG in Chinese children.

Turning now to the effect of language in each reading group, differences between languages were mainly observed between the groups of poor readers: Chinese children showed larger activations than French in the MFG (BF10 = 309.65) and pSTG (BF10 = 74.01) although the same tendency was present in typical readers. The above Bayes factor stayed the same for a wide range of prior specifications (Cauchy prior width: 0–1.5) in the sensitivity analysis. However, there was no evidence for a significant interaction language × reading ability in all these ROIs. On the contrary, the null effect was supported by moderate evidence in the FFG, MFG, STS, PCG, and SPL (respectively BF10 = 0.31, 0.31, 0.31, 0.27, and 0.32, i.e. the likelihood of the null hypothesis BF01 = 3.23, 3.23, 3.23, 3.70, and 3.13); there was no evidence in either direction in pSTG (BF10 = 1.05) (*Table 4*). Sensitivity analysis revealed that the Bayes factor stayed about the same for a wide range of prior specifications (Cauchy prior width: 0–1.5).

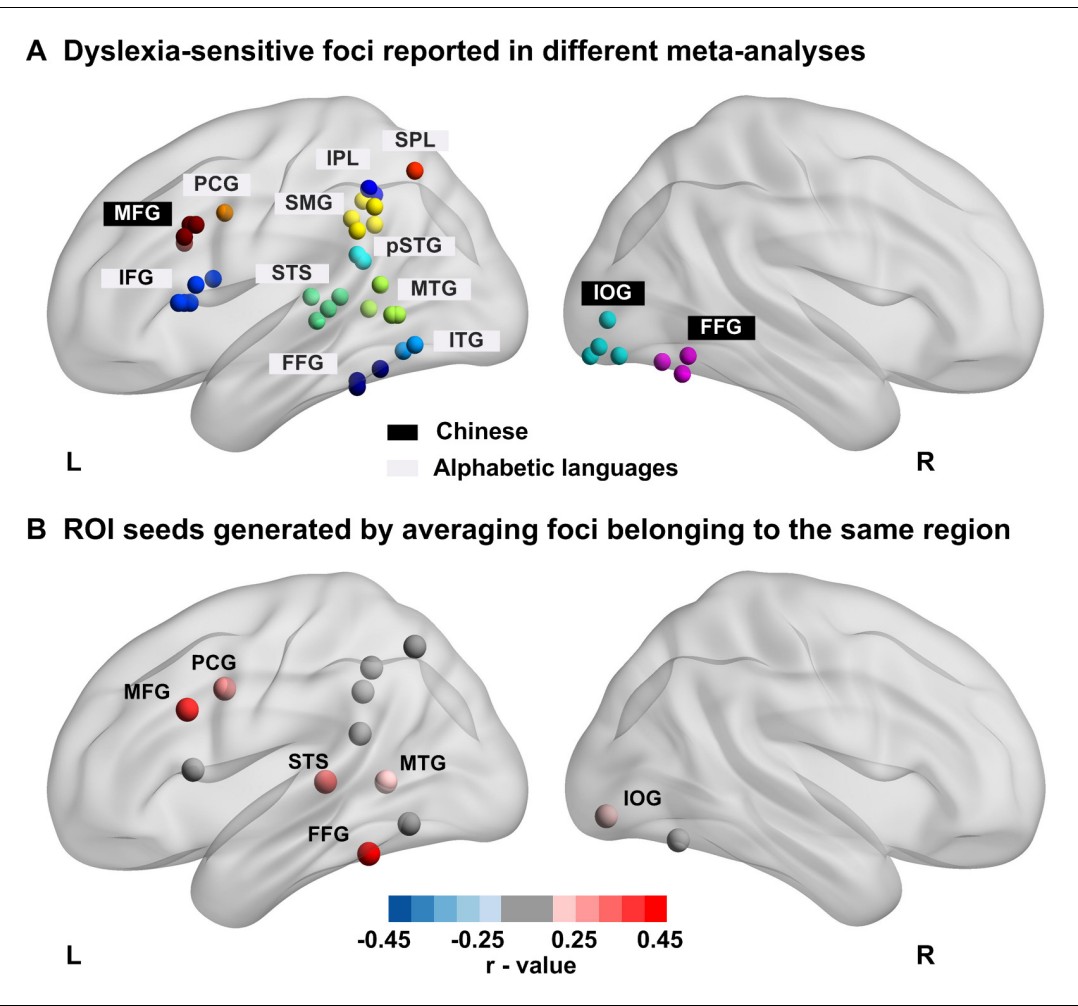

**Figure 3.** Regions of interest (ROIs) used to analyze the data. (**A**) Each sphere represents a peak reported in the literature; Labels in white background indicate foci reported in meta-analyses of dyslexia in alphabetic languages; Labels in black background indicate foci reported in meta-analyses of Chinese reading. (**B**) ROIs used in the current study. Coordinates of foci (see the upper graph) belonging to the same functional region were averaged to create 6-mm-radius spheres at the averaged coordinates. Dots are colored according to their correlation with reading scores across all children. Red dots represent ROIs whose activation to words versus fixation were positively correlated with individual children's reading scores, while gray dots represent non-significant ROIs ($p_{FDR} < 0.05$). MFG: Middle Frontal Gyrus, PCG: Pre-Central Gyrus, STS: Superior Temporal Sulcus, MTG, Middle Temporal Gyrus, FFG: Fusiform gyrus, IOG: Inferior Occipital Gyrus.

These analyses were replicated for the activation to houses and to faces, no main effect nor inter-actions were found in any of these ROIs.

To summarize, when analyses were focused on specific ROIs outlined in the literature as sensitive to reading performance or to differences between writing systems, we replicated the reduced acti-vation to words in poor readers relative to typical readers in the FFG, MFG, STS, and PCG. Crucially, this reduction was observed in both French and Chinese participants with no significant interaction language × reading ability. Activation in SPL, pSTG, and MFG was also modulated by language, with greater activation in Chinese than in French children. For the pSTG, the language effect was mainly observed in the poor readers, due to a large reduction in activation in French poor readers than Chinese poor readers.

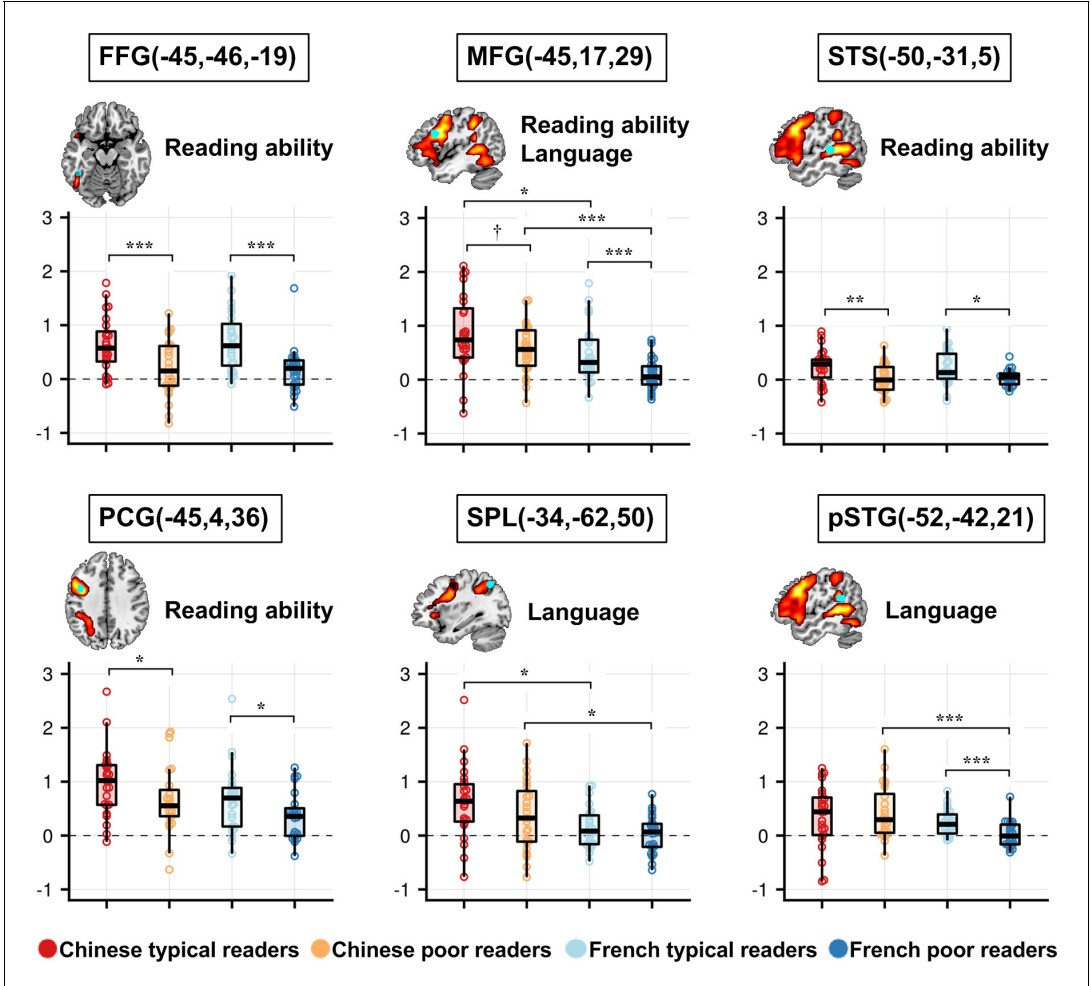

**Figure 4.** Effects of Reading ability and Language on the words versus fixation contrast in the selected ROIs. Brain slices showed the literature-based ROIs (cyan) overlaid on the reading circuit (red-yellow) in our participants (Words > [Faces, Houses]). Plots show the mean activation for words > fixation, in each of the four groups and ROIs. The words 'Reading ability' and 'Language' indicate a significant main effect of reading ability and a main effect of language in the ANOVA (after FDR correction for a total of 13 ROIs). Note that no ROI showed a significant interaction of language × reading ability. Brackets indicate significant post-hoc analyses: **p<0.005, **<0.01, *<0.05, †=0.052.

The online version of this article includes the following source data and figure supplement(s) for figure 4:

**Source data 1.** Activation to words in each of the four groups and ROIs.
**Figure supplement 1.** Profile of activation to words (relative to fixation) in ROIs where the ANOVA did not reveal a significant effect.

## Individual analyses

The above analyses were carried out in a standardized way at the group level. It is therefore possible that the observed group differences were due to a greater inter-individual variability in brain localization in the reading-impaired group than in the typical group. This possibility would lead to a completely different interpretation of the results: each child might have a well-organized brain activity for reading, with the only anomaly being a greater anatomical dispersion in the group of poor readers compared to the group of typical readers. To test this possibility, we performed two individual-based analyses, one based on the comparison of the location and activation values of the most responding voxels and the other examining the stability of the pattern of responses across runs through a multi-voxel pattern analysis (MVPA). We focused on the ROIs previously showing significant differences due to reading ability (i.e. left FFG, MFG, PCG, and STS) and Language (i.e. left MFG, SPL, and pSTG).

**Table 4.** Bayes factor (BF10) in Bayesian ANOVA analyses of children's activation to words versus fixation.

| ROIs | Main effect of language | Main effect of reading ability | Interaction | Post-hoc analyses | | | |
| | | | | Typical readers (Chinese vs. French) | Poor readers (Chinese vs. French) | Chinese children (typical vs. poor readers) | French children (typical vs. poor readers) |
|---|---|---|---|---|---|---|---|
| FFG | 0.216 | 1528.966 | 0.313 | 0.227 | 0.337 | 13.577 | 93.163 |
| MFG | 80.211 | 9.948 | 0.305 | 2.837 | 309.651 | 1.616 | 29.351 |
| STS | 0.220 | 37.507 | 0.309 | 0.387 | 0.259 | 6.933 | 5.076 |
| PCG | 1.532 | 5.625 | 0.275 | 1.782 | 1.075 | 2.662 | 2.661 |
| SPL | 7.052 | 0.562 | 0.323 | 4.486 | 1.782 | 0.822 | 0.332 |
| pSTG | 9.908 | 0.289 | 1.054 | 0.616 | 74.014 | 0.215 | 23.744 |

The value of Bayes factor BF10 means that data are n times more likely under alternative hypothesis (H1) than null hypothesis (H0). The alternative hypothesis in comparisons between typical and poor readers is group 1 (typical) > group2 (poor); the alternative hypothesis in comparisons between languages is group 1 (Chinese) > group2 (French).

## Peak analyses

Considering the locations of the individual centers of mass for word activations in the left FFG, MFG, STS, PCG, SPL, pSTG, their Euclidean distance to the group peaks did not differ between poor and typical readers (*Supplementary file 6*), suggesting a similar dispersion among poor readers and typical readers. These results of a null effect in peak location were further confirmed by Bayesian analyses (the BF10 for an effect of reading ability were 0.22, 0.25, 0.23, 0.36, 0.23, 0.24 for left FFG, MFG, STS, PCG, SPL, and pSTG, respectively, thus supporting the null hypothesis). Furthermore, even after having selected the best responding voxels in each child, the word activation remained weaker in poor than typical readers in the left FFG, MFG, STS, PCG, and pSTG (all $p_{FDR\_corr} < 0.05$). French also yielded weaker activations than Chinese children in the left MFG, pSTG and SPL (all $p_{FDR\_corr} < 0.05$). There was no significant Language × Reading ability interaction in any of these analyses, supported also by the following BF10 in Bayesian analyses (peak location: BF10 = 0.38, 0.30, 0.70, 0.35, 0.31, 0.35; peak activation: BF10 = 0.35, 0.38, 0.59, 0.35, 0.00, 0.53 for left FFG, MFG, STS, PCG, SPL, and pSTG). These results thus corroborated the standard analyses.

As concerns faces, we did not observe any effect of reading ability or Language on the peak locations and activations in bilateral fusiform face gyrus (FFA) in both frequentist (all p>0.09) and Bayesian statistics (all BF10 < 1).

## MVPA of the activations to words

The previous analysis asked whether, at the individual level, the peak activity is reduced in poor readers. In the present section, we ask the same question about the stability of the response pattern across runs (i.e. the within-subject reliability of the activation patterns). Indeed, the reduced activation in the group analysis could be due to two distinct causes. Poor readers could have a genuinely reduced and erratic activation, but alternatively, poor readers could possess an identifiable and reproducible circuit similar to typical readers, only with a spatially more dispersed extent. In that case, the within-subject reproducibility of multivariate activation patterns may not differ between typical and poor readers.

To examine the within-subject reliability of activation across runs, we used a multi-voxel pattern analysis (MVPA) focused on the same regions than above. We computed the correlation between pattern of activations in run 1 and 2, separately for within-category patterns (words in run 1 and words in run 2) versus between-category patterns (average of words-faces, words-houses and faces-houses, each in run 1 versus run 2). A reproducible reading circuit should result in a significant effect of condition (greater within-category correlation than between-category correlation). Furthermore, if the activation pattern is less reproducible in poor readers than in typical readers, there should be a significant interaction of this Condition effect with Reading ability.

In all the ROIs, when pooling over all subjects, there was an overall replicable pattern of activation evoked by words, as indicated by a significant main effect of condition, with a greater correlation coefficient within words than between words and other categories (all $p_{FDR\_corr} < 0.001$) (see *Figure 5*

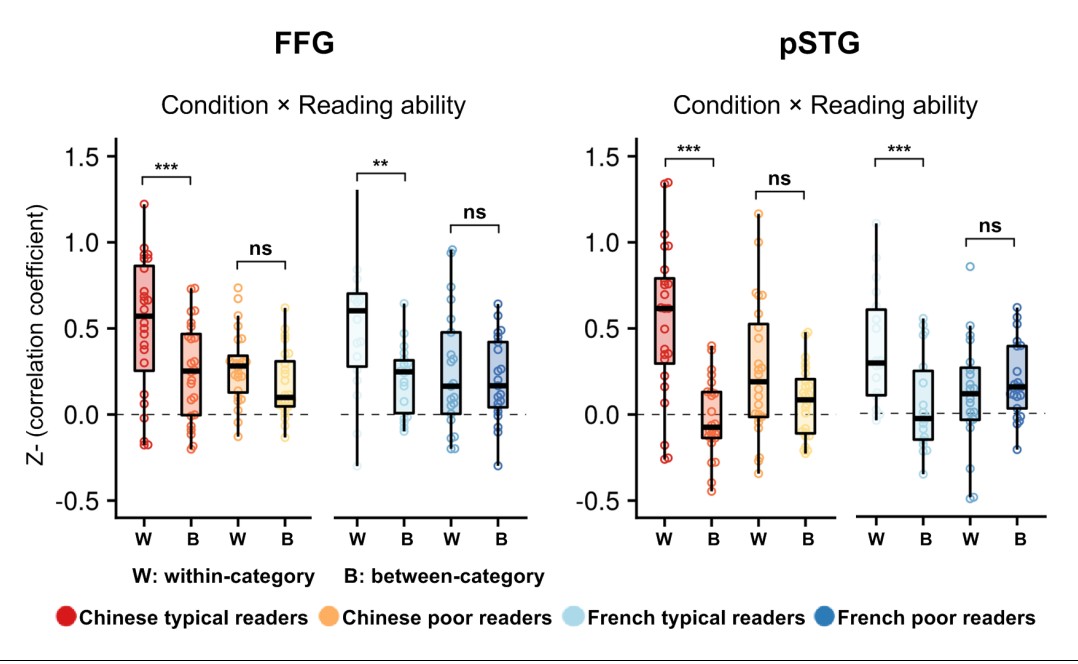

**Figure 5.** Multivariate pattern analysis indicating that the word-induced activation is less reproducible in poor readers. Within the designated ROIs, we computed the correlation coefficient of the multivoxel patterns for the word versus fixation contrast in run 1 and in run 2 (within-category correlation). For the between-category coefficient, the plots show the average correlation coefficient between words and faces, words and houses and faces and houses between run 1 and run 2. In each plot, the correlation is presented for the Chinese children on the left of the plot and for French children on the right. The words 'Condition × Reading ability' indicate a significant interaction between condition (within vs. between) and the status of children (typical vs. poor readers) (after FDR correction for a total of 13 ROIs). Typical readers, but not poor readers, exhibited a significant similar pattern of activation to words from one run to the next in the left FFG and pSTG, suggesting that the weaker activation for words in poor readers was not due to greater anatomical variability in this group but was related to genuinely less reproducible activation patterns than in typical readers.

The online version of this article includes the following source data and figure supplement(s) for figure 5:

**Source data 1.** Correlation coefficients for within-category patterns and between-category patterns in each of the four groups and ROIs.

**Figure supplement 1.** MVPA analyses.

---

and *Figure 5—figure supplement 1A*). Crucially, we also observed a significant interaction of condition × reading ability (typical vs. poor readers) in the left FFG (F (1, 83)=10.14, p=0.002, $p_{FDR\_corr}$ = 0.006) and in the left pSTG (F (1, 83)=15.75, p<0.001, $p_{FDR\_corr}$ < 0.001). Post-hoc analysis found that typical readers, but not poor readers, exhibited a significantly similar pattern of activation from one run to the next. Those results show that the above differences between typical and poor readers were not due to an artifact of group averaging, and that children struggling with reading acquisition exhibited a genuinely more erratic activation pattern for words in these regions (*Figure 5*).

## MVPA of the activations to faces

For the MVPA analysis in the bilateral fusiform face areas (bilateral FFA), only the main effect of condition reached significance ($p_{FDR\_corr}$ < 0.001), with a greater correlation coefficient for within-category patterns (faces-faces) than for between-category patterns (faces-words, faces-houses, words-houses). Neither the main effect of reading ability nor the interaction of condition × reading ability reached significance. These results suggest an equally replicable pattern of activation to faces in poor and typical readers bilaterally in the fusiform face area (see *Figure 5—figure supplement 1B*). This pattern resulted in a triple interaction of category (words versus faces)×condition × reading

ability when the fusiform data from words and faces were analyzed together, indicating that the reduced stability of activations in poor readers was limited to words.

In summary, individual analyses substantiated a genuine reduction and instability of the activation to words in poor readers relative to typical readers. Importantly, this instability was not universally present in all visual categories (as might be the case if, for instance, the poor readers had greater noise or motion), because the activations to Faces in the fusiform regions did not differ between groups.

### Anterior-to-posterior gradient in the visual cortex

Finally, given the massive impact of reading on the organization of the ventral temporal areas, we focused on these regions and examined the anterior-posterior gradient of responses for the different visual categories vs. fixation.

### Words

Keeping constant x = ±48 and z = −16, we studied the activation to Words vs. fixation along the y-axis (ranging from −79 to −22). Firstly, we observed greater response to words in Chinese children compared to French children in the right hemisphere at several y coordinates, leading to a significant triple interaction of language × hemisphere × ROI. Secondly, we observed larger activation in the posterior relative to anterior sites as revealed by the main effect of ROIs. Crucially, the reading ability × ROIs interaction was significant. In more detail, compared to typical readers, poor readers had weaker activation to words at several consecutive sites (y-axis at −64, −55, −46, and −37). However, when we considered separately Chinese and French children, only one site (y = −46) survived correction for multiple comparison in both Chinese and French children. This site is only slightly anterior to the classic VWFA site (*Cohen et al., 2002*) (see *Figure 6B*).

### Faces

Keeping constant x = ±39 and z = −16, we also studied the activation to Faces vs. fixation along the y-axis (ranging from −79 to −22). We observed a significant Hemisphere × ROI interaction. This effect was due to greater right than left face activation at several y coordinates (y = −73, −64, −55, −46). Besides, the main effect of reading ability reached significance, due to a lower activation to faces in poor readers compared to typical readers bilaterally and in both languages (*Figure 6C*).

### Houses

Along the medial house-specific activation at x = ±30 and z = −16, we similarly studied the activation to Houses vs. fixation. We found a significant triple interaction of language × hemisphere × ROI. French children had greater right than left activation at each of the six anterior-posterior y coordinates (all $p_{FDR\_corr} < 0.005$) while Chinese children had the same pattern only at four sites (y = −73, −46, 37, 28). We also observed a significant reading ability × ROI interaction, with decreased activation to Houses in poor readers in several sites (*Figure 6D*).

In summary, we recovered the expected anterior-posterior and medial to lateral gradient of preferences for the different visual categories. Chinese children had larger right-hemispheric activation to words than French children; and poor readers, in both languages, showed lower activation than typical readers at the VWFA site. Congruent with the whole-brain analyses, we observed an effect of reading ability on Face activations in both hemispheres. The activation to Houses was also reduced in poor readers at several sites.

## Discussion

In the present study, we examined the universality and specificity of the neural bases of reading among novice readers by comparing two very different writing systems: French and Chinese. Our goal was to study whether the same neural circuit was involved in the success and failure of reading acquisition, regardless of the complexity of the written symbols and the size of the speech units that are mapped onto them. We investigated this question in 96 10-year-old children using a similar paradigm in both countries with a minimally demanding task (i.e. detecting a star) that did not directly

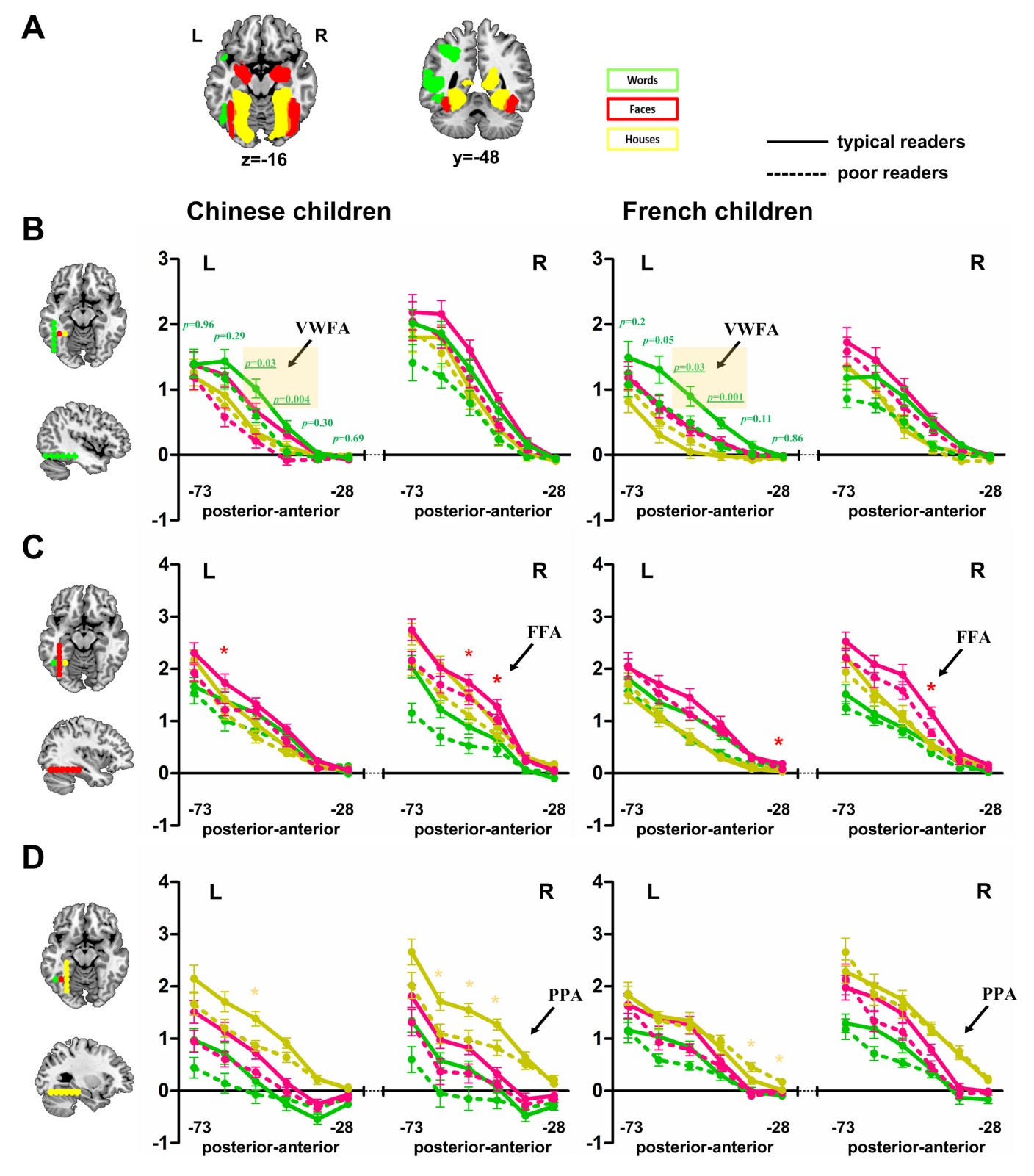

**Figure 6.** Mosaic of preferences for different visual categories (Words, Faces and Houses) in the ventral visual cortex. (**A**) Slices show the activation difference between a given category and the others across all participants. (**B**) fMRI signal change of Words relative to fixation in both Chinese and French children in successive cortical sites along the y-axis (green dots on the left cortical slices) with constant x = ±48 and z = −16. Both Chinese and French poor readers have significantly lower activations relative to their controls at a specific y site of y = −46 ($p_{FDR\_corr}$ = 0.048 and $p_{FDR\_corr}$ = 0.012

*Figure 6 continued on next page*

*Figure 6 continued*

respectively for Chinese and French children) corresponding to the classical coordinates of the VWFA (*Cohen et al., 2002*). (**C**) fMRI signal change of Faces relative to fixation in both Chinese and French children in successive cortical sites along the y-axis (red dots on the left cortical slices) with constant x = ±39 and z = −16. (**D**) fMRI signal change of Houses relative to fixation in both Chinese and French children in successive cortical sites along the y-axis (yellow dots on the left cortical slices) with constant x = ±30 and z = −16. Additional files.

The online version of this article includes the following source data for figure 6:

**Source data 1.** Activaions to different categories relative to fixation in ROIs in the ventral visual cortex.

tax reading or reading-related cognitive skills and thus was equally easy for everyone in four age-matched groups (French and Chinese × typical and poor readers).

First, in a whole-brain analysis, we recovered the classical category-specific activations for words, faces and houses in extra-striate visual areas across all participants but also in each group (*Figure 1*). Second, reading scores were correlated with the word activations in common key regions of the reading circuit (left VWFA, posterior superior temporal gyrus/sulcus, middle frontal gyrus and precentral gyrus) but also in the right hemisphere (middle occipital and fusiform gyri and precentral). Third, analyses based on ROIs from the literature confirmed these results, including an effect of reading ability in the left middle frontal gyrus which was true in both French and Chinese children, contrary to previous suggestions of a Chinese-specific effect at this site. The main effect of reading ability across languages was replicated even when analyzing only the most responsive voxels in each individual.

We also observed a few differences in activations depending on the children's native language. Chinese reading tended to engage more symmetrical activations in the visual system, with stronger activations in the right hemisphere than French readers when we specifically tested the anterior-posterior organization of the fusiform region. Chinese children also had stronger activations than French children in the left parietal region, middle frontal region and posterior superior temporal gyrus.

We concluded our analyses by examining the reproducibility of the activation patterns between runs. The within-subject pattern of activity evoked by words was reproducible across runs in typical readers in all key reading regions, underscoring that the reading circuit is stable after 3 years of learning to read and can be reliably measured in a single fMRI run even in children. However, such was not the case for poor readers, whose activity was significantly less reliable in left fusiform and posterior superior temporal gyrus in both Chinese and French poor readers. We now discuss each of these results in turn.

## The reading network is largely universal, and modulated by reading skill

A long-standing debate in reading research is whether the neurobiological circuitry for reading is universal across languages. Previous cross-cultural fMRI studies have compared brain activations in adult readers in different languages and suggested that the expert reading network may be universal across languages (*Nakamura et al., 2012*; *Paulesu et al., 2000*; *Rueckl et al., 2015*). The current study extends this finding to young children by showing common activation patterns to words in children in both alphabetic and non-alphabetic writing systems after only 3–4 years of primary school. Script invariance across English and Chinese was already reported after only one year of reading acquisition in the VWFA (*Krafnick et al., 2016*). This region is also the most strongly modulated by reading proficiency in both French and Chinese, with no difference (Bayes factor < 1) between French and Chinese typical readers but also between French and Chinese poor readers, thus confirming that VWFA activation is a universal marker of reading proficiency. The same observation can be made to a lesser extent for the STS and the PCG, the activation of which also depends on the reading proficiency in both writing systems in a similar way.

These cultural invariance findings are in agreement with the neuronal recycling hypothesis, according to which recent cultural acquisitions (e.g. reading) rely on the preemption of universal pre-existing circuits of the human brain, with only small culture-dependent modulations (*Dehaene and Cohen, 2007*). In any language, reading recruits a pre-existing circuit that connects visual areas capable of learning to recognize orthographic symbols, and spoken language processing areas (*Hannagan et al., 2015*; *Saygin et al., 2016*). Successful literacy acquisition is thus the result of the convergence of visual and speech processing systems, both of which are likely to be largely universal

and partially laid down under genetic control. Indeed, the spoken language network is already present at its usual left-hemispheric location in 2-month-old babies (*Dehaene-Lambertz et al., 2002*; *Dehaene-Lambertz et al., 2006*; *Dehaene-Lambertz and Spelke, 2015*). The specific subpart of the ventral visual pathway which is used to recognize written characters seems to be, at least in part, determined by its pre-existing connections with this language network (*Barttfeld et al., 2018*; *Bouhali et al., 2014*; *Hannagan et al., 2015*; *Saygin et al., 2016*) and by its micro-structure (*Weiner et al., 2016*). Thus, according to the neuronal recycling hypothesis, in spite of variations in language and writing systems, a considerable amount of inter-cultural convergence should be expected.

In our study, although the main areas for reading were common to both groups, we also observed modulations of the amplitude of brain activity within culturally universal brain circuits. Chinese children had larger activations than French children in (1) left intraparietal sulcus; (2) left middle frontal gyrus (BA 9); (3) right hemisphere occipitotemporal regions; and (4) left posterior superior temporal sulcus.

Brain imaging research has well documented the role of the left intraparietal sulcus in visuo-spatial encoding and attention (*Davranche et al., 2011*; *Offen et al., 2010*), including the serial analysis of letters in a word or pseudoword (*Cohen et al., 2008*). We therefore speculate that the greater involvement of this region in Chinese reading is presumably due to the greater amount of spatial analysis required by Chinese character recognition. Indeed, activations in the inferior parietal region are observed in French children (*Dehaene-Lambertz et al., 2018*), but only transiently during initial reading instruction, when children must effortfully pay attention to the left-right succession of letters and disambiguate mirror-letters. Congruently, improvement in reading skills over the first year of school correlates with microstructural changes in the connectivity between the VWFA and the inferior parietal region (*Moulton et al., 2019*). Activation in the intraparietal sulcus is also needed when adults read in an unusual format (vertical French words for example) (*Cohen et al., 2008*) or determine the position of a letter in a word (*Ossmy et al., 2014*), underscoring the role of this region in orienting attention and computing the spatial relationships of word elements. Our results indicate that this process is involved in alphabetic languages as well as in Chinese, but its activation may remain weak in alphabetic languages due to the simplicity of the spatial relationships.

The greater involvement of the left middle frontal gyrus (BA9) in Chinese than in French children is in line with existing research (*Bolger et al., 2005*; *Tan et al., 2005a*). This region has been proposed to support language-specific processing required by Chinese reading, such as whole-syllable retrieval (*Tan et al., 2005a*), the tonal nature of Chinese phonology (*Gandour et al., 2002*), memory-based lexical integration of orthography, phonology and semantics (*Perfetti et al., 2005*), visuo-spatial working memory (*Wu et al., 2012*) or writing gesture information (*Cao and Perfetti, 2016*; *Nakamura et al., 2012*). In the reading of English, its proposed functional role is also diverse: lexical semantics (*Bolger et al., 2005*), phonological processing (*Pugh et al., 1996*), lexical selection (*Bolger et al., 2008*), or grapheme–phonology conversion (*Jobard et al., 2003*).

Perhaps the most likely explanation is that the LMFG, as well as the more medial and dorsal Exner area, are systematically involved in writing (*Planton et al., 2013*; *Purcell et al., 2011*). FMRI studies in typical adult readers, comparing French and Chinese reading, have shown that the left MFG is not specific to Chinese writing, but includes a representation of handwriting gestures that is engaged in both alphabetical and non-alphabetical languages (*Nakamura et al., 2012*). As proposed by *Nakamura et al., 2012*, the activation in the MFG and PCG, as seen here, might correspond to the 'reading by hand' circuit (gesture recognition system) as opposed to the 'reading by eye' circuit (shape recognition system in the VWFA). Furthermore, writing is not only learned in parallel with reading, but facilitates reading acquisition in children (*Bara et al., 2004*). Thus, the involvement of the left MFG in French children, although less important than in Chinese children, might be due to a heavier dependence on handwriting during reading acquisition than during adulthood. This interpretation is supported by a recent study showing that, in second-language learners of Chinese, viewing characters that were learned through character writing induced greater activation in LMFG compared with those learned without character writing (*Cao and Perfetti, 2016*).

The greater activation in the right ventral visual system in Chinese reading has been reported in previous literature and this region has been found to show a children-to-adult developmental increase for Chinese, suggesting that this region is especially important in Chinese reading

acquisition, presumably because of its involvement of holistic visuo-orthographic processes (*Bolger et al., 2005*; *Cao et al., 2014*; *Tan et al., 2005a*).

We also found that Chinese children showed slightly larger activation than French children in the posterior superior temporal gyrus/sulcus. This region is associated with phonological processing and grapheme-phoneme conversion (*Turkeltaub et al., 2003*). Previous findings in adults suggested that it is more engaged in alphabetic than in logographic languages (*Bolger et al., 2005*; *Tan et al., 2005a*), but our study showed an opposite pattern. In Chinese reading, a large number of written characters correspond to the same syllable, thus phonological information is insufficient to access semantics of a printed character. As a result, Chinese readers must rely more heavily on the direct route from orthography to the lexicon (*Cao et al., 2010*; *Shu et al., 2003*). However, phoneme-level representation still plays an essential role in learning to read Chinese, all the more that syllable decoding is complex. This may be especially true in beginners, who rely on Pinyin, an alphabetic notation which allows children to manipulate different phonological units such as decomposing a syllable into onset, rime, tone and phonemes. Greater dependence on the grapho-phonological route in children than in adults persists until at least grade four in Portuguese (*Ventura et al., 2007*). It is also possible that the need to suppress the activation of the syllable associated with the character that competes with character identification, transitorily increases activation in that region. Furthermore, when carefully comparing the activation of the posterior temporal gyrus with other regions, we found that the greater activation in Chinese was mainly due to a weak activation in French poor readers (see *Figure 4*). These children were defined as dyslexics in France and might have been more impaired than the Chinese poor readers. Interpretation must be very cautious, given that the language × reading ability interaction was not significant, and that the Bayesian analysis provided no evidence favoring H0 or H1. Further research will be needed to probe whether a reduction in pSTG activation, presumably due to a phonological impairment, is a more frequent cause of reading disability in alphabetical scripts than in Chinese, as suggested by numerous prior studies of dyslexia (*Hulme et al., 2002*; *Ramus and Szenkovits, 2008*; *Wydell and Butterworth, 1999*).

Overall, our findings indicate that learning to read largely involves the same key regions across cultures and ages, but with quantitative modulations depending on the specific demands of the task, the learning stage, and culture-dependent characteristics. Our novice readers, who probably needed to deploy all their resources in order to succeed in reading, may be more informative than expert adults in objectifying the entire reading circuit, unless the latter are pushed to their limits by unusual, thus less automatized, format of words presentation, as was done for instance by *Cohen et al., 2008*.

## A universal neural phenotype for reading disability

Some authors have also proposed that the cortical regions mediating reading disability are different in Chinese and alphabetic languages (left middle frontal gyrus in Chinese versus left temporo-parietal regions in alphabetic languages) (*Siok et al., 2008*; *Siok et al., 2004*). On the contrary, our results were strikingly similar in Chinese and French poor readers tested with the same paradigm, thus suggesting a universal neural phenotype for reading disability. Those children were impaired in all classical reading-related regions, most notably the VWFA in the left FFG and the left posterior STG, regions that have been consistently reported to show lower activations in dyslexics relative to controls in alphabetic languages (*Blau et al., 2009*; *Martin et al., 2016*; *van der Mark et al., 2009*). The present study confirms that these reductions in activation can be observed since childhood in all writing systems, whether alphabetic systems with deep or shallow orthographies, or in Chinese characters. *Hu et al., 2010* reached a similar conclusion at a later age by comparing a smaller group of 11 Chinese and 11 English dyslexics.

Thanks to individual peak location and intensity analyses, as well as multivariate pattern analyses, we could reject an alternative interpretation which, to the best of our knowledge, was not explicitly tested in previous studies: the possibility that the reduced activations are an artifact of group averaging, solely due to greater inter-individual variability in the localization of reading-related circuits in the dyslexic brain. Using individual peak, we observed that the brain localization to words were not more dispersed among poor readers than among typical readers. Using MVPA, we showed that, within individual subjects, the activation patterns in the VWFA in response to written words were less reproducible across runs in poor readers than in typical readers. This was solely the case for words, not for the other visual categories. We did observe a slightly reduced activation to faces and

houses in poor readers relative to typical readers, as previously reported in illiterate subjects (*Dehaene et al., 2010*), as well as seen in response to non-word stimuli (numbers, abstract strings) (*Boros et al., 2016*) and faces in dyslexics (*Gabay et al., 2017*; *Monzalvo et al., 2012*). However, the pattern of activity for faces was stable from one run to the next, contrary to what was found for words in the MVPA analyses. This observation suggests that the reduced activation to written words may not reflect a general disorganization of the extra-striate visual areas, but rather a specific difficulty with written words.

Given that the site of the VWFA does not activate strongly to written words in illiterates (*Dehaene et al., 2010*; *Hervais-Adelman et al., 2019*) and in typical children before they learn to read (*Dehaene-Lambertz et al., 2018*), its under-activation in poor readers may simply reflect the lack of reading practice – in other words, it might be a consequence rather than a cause of the reading deficit. Dyslexia is likely to be a heterogeneous deficit with a variety of different causes, including a phonological deficit in many children, but also visual attentional deficits, plausibly anywhere along the complex processing chain that leads from print to sound and meaning (*Friedmann et al., 2010*; *Paulesu et al., 2014*; *Valdois et al., 2012*). The present study cannot distinguish cause from consequence, but it does demonstrate that a reduced activation in the VWFA is one of the dominant fMRI signatures of reading poor performances.

The other major site of atypical activation reported in the dyslexia literature is the left posterior STG, a region involved in grapheme-phoneme conversion (*Frost et al., 2009*; *Turkeltaub et al., 2003*) although this location is less consistently found in children than in adults (*Richlan et al., 2011*). In our data, only French poor readers had a significantly lower activation relative to their controls at this location, but in both languages, poor readers exhibited an unstable activation pattern across fMRI runs, suggesting fragile phonetic representations evoked by written stimuli. In agreement with this interpretation, *Vandermosten et al., 2020* also observed disrupted phonological representation in a multivariate pattern analysis but not in the univariate activation analysis in beginning readers with a family risk for dyslexia. Again, because this site is also under-activated during spoken language processing in illiterates and in preliterate children compared to adult and children readers (*Dehaene et al., 2010*) we cannot ascertain whether its anomalous activation is a cause or a consequence of the reading disability. Indeed, it does not have to be one or the other, but could be both, as phonological awareness is known to be both a predictor of future reading (*Hulme et al., 2012*; *Melby-Lervåg et al., 2012*), and a consequence of learning to read in an alphabetical language (*Morais et al., 1986*).

Finally, we also observed decreased activation in the left middle frontal gyrus, as first reported in Chinese dyslexics by Siok and collaborators (*Siok et al., 2008*; *Siok et al., 2004*). Activation at this location was modulated by both reading score and language: overall, Chinese children had larger activation than French children at this site, but in both languages, poor readers also exhibited weaker activation than their controls. Thus, the effects of reading ability and language on that region seemed to be additive. The first reports of dysfunction in this area were obtained in Chinese dyslexics and, because of a lack of evidence for a comparable deficit in alphabetic languages, were interpreted as supporting a unique contribution of this region to Chinese reading (*Siok et al., 2008*; *Siok et al., 2004*). Here, however, we observed reduced activations in poor readers relative to typical readers in Chinese, but also in French. The language × reading ability interaction was not significant, and Bayesian analyses confirmed the moderate evidence for a null interaction. Those Bayesian analyses, together with the fact that we easily detected strong differences between groups (mainly related to reading ability, but also, in a few regions, to the children's native language) support the hypothesis that our study, with 96 subjects, was sufficiently powered and that cross-linguistic differences in neural deficits in poor readers were indeed minimal or absent, in this region as well as elsewhere in the brain. While a null effect can never be firmly asserted, the present results positively indicate that atypical activation in this region is not unique to Chinese poor readers, but can be found in alphabetic readers as well and is therefore part of a universal phenotype of reading disability. Note that these results are congruent with our previous study of French children, where LMFG exhibited significant activation to words relative to other visual categories only in typical readers, not in dyslexics (*Monzalvo et al., 2012*).

Unlike readers of alphabetic languages who only have to acquire a small set of letters and of grapheme-phoneme correspondences, readers of Chinese must learn a few thousands of characters. To overcome this difficulty, Chinese readers may rely more on a motor memory of writing gestures

as a means for memorizing the large number of characters, and as noted earlier, writing skills predict reading ability in Chinese children (*Tan et al., 2005b*). The greater activation of the left MFG may therefore reflect the greater reliance of Chinese children on writing, and the dysfunction of this region in Chinese poor readers may reflect an underlying deficit in memorizing writing gestures (*Ziegler, 2006*). This strategy is not unique to Chinese, however: novice readers in alphabetic languages also benefit from a motor memory for hand gestures when recognizing written words (*Bara et al., 2004*), and a recent study found that a motor representation of handwriting gestures is automatically accessed for subliminal written words in both Chinese and French adult readers (*Nakamura et al., 2012*). Our findings on children further suggest that the left MFG is likely to play a pivotal role in successful reading acquisition independently of the writing system.

## Limitations and future direction

Several limitations of the present study need to be considered. First, our task was orthogonal to reading, and involved the mere detection of a picture of a star. This choice was made to avoid any effect due to performance itself. A task that is more difficult for some children than from others could have induced greater activation, but also greater movement in the scanner. The disadvantage is that participants were not explicitly instructed to attend to words, which could have increased the differences between poor and typical readers, and between French and Chinese readers. Note, however, that our results indicate a strong sensitivity of our procedure to reading proficiency, since reading performance was correlated with activation in all key areas of the reading circuit (VWFA, posterior STG, MFG, and pre-central (*Figure 2* and *Table 3*)).

Although our groups were larger than in any previous study comparing Chinese and alphabetic writing systems, and were also more homogeneous in age, whole-brain analyses at corrected-level of significance failed to show significant differences between poor and typical readers. Our conclusions are therefore based primarily on analyses of ROIs that were previously established in the literature as being critical to reading proficiency. Our small effect size may be due to the definition of our groups, which included readers with poor reading performances but not all full-blown dyslexics. Indeed, the reading delay in Chinese children was less than in French children (*Table 1*), but even in French children who were typical French dyslexics (more than 2 years of delay in the LUM test), whole-brain analyses were poorly sensitive, which underlines the difficulties of pediatric research. Constraints on experiment duration are severe with young children, and make it difficult to collect the same extensive datasets as in adults. A lack of sensitivity could also have arisen from the greater variance between children in terms of maturation, learning experience, attention, and concentration on the task compared to adults who can remain attentive, perform more consistently, and are generally at the peak of the skill being tested (i.e., no longer learning). Furthermore, the difference between impaired readers and typical readers may be smaller in children than in adults because all children are in the process of learning and are therefore not fully competent, whereas the adult typical readers (often recruited at university) are at ceiling. Finally, as has been shown in alphabetic languages, the slope of normal reading acquisition in children depends on the transparency of the writing system. In other words, a typical 10-year-old reader would not perform equally well in English, French, German or Italian (even if we could use exactly the same test) because he or she has not yet attained the same level of reading fluency (see *Ziegler et al., 2003* for a comparison between German and English). Thus, comparing languages and assessing equivalent reading delay in different writing systems, while still matching for age and education, raises complex issues and increases the difficulties of intercultural comparisons in children compared to adults.

However, since we had a particular interest in understanding the role of previously identified brain regions, an analysis based on ROIs was a reasonable choice, having the great advantage of mitigating the problem of multiple comparisons and reducing the risk of false negatives, but also allowing us to examine activation patterns at an individual level, as in the MPVA analyses. These targeted analyses were able to show the role of MFG in alphabetic languages. As usual in science, the results will need to be confirmed in future studies using larger samples to obtain more robust inferences.

In conclusion, with several convergent analyses, we revealed that the neural bases of reading in typical children and in those struggling with reading acquisition are largely similar, but partly language-specific, in French and Chinese readers. Across these very different writing systems, the cultural invention of reading relies on similar brain resources. As previously noted in an adult fMRI

study (*Nakamura et al., 2012*), cultural variability is merely reflected in the variable emphasis that different writing systems put on phonemes, syllables and whole words, which in turn may modulate the severity of dyslexia and the degree of anomaly that can be detected at different locations along the brain's reading circuitry.

# Materials and methods

## Participants

Ninety-six children participated in the current study, including 24 Chinese poor readers (mean age = 123 months, standard deviation (SD) = 10), 24 Chinese typical readers (mean age = 123 months, SD = 11), 24 French poor readers (mean age = 123 months, SD = 10) and 24 French typical readers (mean age = 123 months, SD = 11). All children reported normal hearing and corrected-to-normal vision and no history of neurological or psychiatric disorder. Nonverbal IQ, assessed by Raven's Standard Progressive Matrices in Chinese children and Wechsler Intelligence Scale in French children, was in the normal range for all participants. The study was approved by local institutional review boards in Beijing (China) and Kremlin-Bicêtre (France), respectively. Written consent was obtained from all children and their parents.

## Chinese participants

Because standardized tests of dyslexia are not available in Chinese, we tested a large population of 2554 primary school children in Beijing (3rd grade- 5th grade, 10- 13 years of age) to calculate the standard norms in the following tests. The first round of tests involved: (1) Chinese Character Recognition Test (CCRT) (*Wang and Tao, 1993*), (2) Reading Comprehension Test (RCT) (*You et al., 2011*), (3) Raven Progressive Matrices Test (*Raven et al., 1994*), and (4) Digit Cancellation Test (*Mirsky et al., 1991*). Children with a CCRT Z-score below −1.25 SD were identified as children with potential reading difficulties. We then invited these children and their parents to take part in a second-round of tests, which involved MRI scanning and several individually-administered tests: (5) Chinese Phonological Awareness Test (CPAT), (6) Character Reading Test (CRT) (*Li et al., 2012*), and (7) Rapid Automatized Naming Test. A total of 103 children with different age and reading abilities were scanned and more information about these children can be found in a previous paper (*Li et al., 2018*). We considered poor readers to be those whose CCRT Z-score, or CPAT Z-score, was consistently low in both rounds of tests (specifically < −1.25 SD in the first and < −1.5 in the second) and obtained 24 children. We then selected an equal number of typically developing children (>−0.5 SD in all reading-related tests) whose age, sex and non-verbal IQ matched those of the reading-struggling group (*Table 1* and *Supplementary file 1*). By definition of the groups, the two groups were significantly different (CCRT: t = 18.03, p<0.001).

## French participants

To match the Chinese children, we selected 24 dyslexic and 24 typical readers from two previously published French studies: 21 dyslexics and 18 typical readers from the Monzalvo et al's study (*Monzalvo et al., 2012*), and 3 dyslexics and six typical readers from the population of *Altarelli et al., 2013* study. Dyslexia was diagnosed in a dedicated learning disability center based on extensive behavioral testing with nationally established criteria following INSERM recommendations (clinical examination, full-scale IQ, standardized tests for working memory, meta-phonology, spelling, rapid automatic naming, etc.). At the time of fMRI scanning, we checked the children current reading level with 'L'alouette', a standardized reading test classically used to detect dyslexia in French speaking children (*Lefavrais, 1967*). It consists in reading as fast and accurately as possible a meaningless text of 265 words within 3 min (*Lefavrais, 1967*). All French poor readers had a delay of 20 months, or more, in this test. French and Chinese children were matched in age and sex in each group (see also *Table 1* and *Supplementary file 2*). By definition of the groups, the two groups were significantly different ('Alouette': t = 14.66, p<0.001).

Due to differences in writing systems, the reading skills expected at a given age are different in alphabetic languages and Chinese. Here, French children were assessed by reading a text and Chinese by character recognition. Therefore, the direct comparison of children's performance across languages provides an indication of the distance between typical readers and the children with

reading difficulties, rather than an absolute assessment of dyslexia. While reading scores were similar in French and Chinese typical readers (t < 1), French poor readers tended to have worse scores than Chinese poor readers (t = 2.45, p=0.07), leading to an interaction between language and group (F (1,92)=3.91, p=0.051).

## Stimuli and task

For French children, the experimental procedure was identical to *Monzalvo et al., 2012*. The procedure was adapted to Chinese children by replacing French words and Caucasian faces by Chinese words and Asian faces. While being scanned, Chinese and French children viewed short blocks of words, faces, and houses and of a revolving checkerboard (30 frequent regular words known by young readers and 30 black and white pictures in each category) followed by a fixation cross during 10.5 s (total bloc duration 28.5 s). In each block, 10 pairs of different images belonging to the same category (200 ms presentation for the first picture/word, 200 ms inter-stimulus, 500 ms presentation for the second picture/word) were presented, separated by a 600 ms fixation period. Besides, two stars were randomly inserted in each block, 1500 ms for each star. Children were instructed to press a button with their right index finger whenever a target star appeared. This task was designed to keep their attention focused on the visual stimuli, but without any explicit reading requirement. For Chinese children and the older French children, a supplementary category (tools) was added but not included in the present analyses as this category was not presented in the original study (*Monzalvo et al., 2012*) and thus missing in most of the French children reanalyzed here.

In each run, there were two blocks of each visual category and only one block of checkerboard. All the blocks were presented in a random order. Chinese children performed two runs and French children performed four runs in *Monzalvo et al., 2012*, and only one run in *Altarelli et al., 2013*.

fMRI Acquisition Parameters fMRI data were acquired on two Siemens 3T scanners using a 12-radiofrequency-channel head coil and the same gradient-echo planar imaging sequence in France and China with the following parameters: 40 contiguous 3 mm isotropic axial slices, TE/TR = 30/2400 ms, flip angle = 81°, voxel size = 3 × 3 × 3 mm, matrix = 64 × 64. A high-resolution T1 weighted volume was also acquired with the following parameters: 176 axial slices, TE/TR = 4.18/2300 ms, flip angle = 9°, matrix = 256 × 256.

Prior to the scanning session, all children underwent a training session in a mock scanner. This training session aimed to help children become familiar with the MRI environment and task instructions, and to teach them to keep their head motionless during the scan.

## Data pre-processing and statistical analyses

Preprocessing and analyses of the data were done using SPM12. The French and Chinese data were processed together. The functional images were first corrected for differences in slice-acquisition time and realigned to the first volume in the scanning session. ArtRepair toolbox was used to detect and repair bad volumes (*Mazaika et al., 2007*). Two criteria were used to detect bad volumes: (1) 1.5% variation in the global average BOLD signal from scan to scan and (2) 0.5 frame-wise displacement, reflecting the sum of all head movements from scan to scan (calculated from realignment parameters). The damaged volumes that exceeded these criteria were replaced by linear interpolation of previous and subsequent images or by nearest-neighbor interpolation when several consecutive images were affected.

For the anatomical image, we first checked for scanner artefacts and gross anatomical abnormalities, then we manually set the origin of T1 image to the anterior commissure for each subject. We normalized each child's anatomy to the Montreal Neurological Institute (MNI) template using the DARTEL approach to improve segmentation accuracy and local registration among participants. Functional images were co-registered to their corresponding anatomy. Then the parameters obtained during the DARTEL wrapping process were applied to the functional images which were finally smoothed using a 6 mm Gaussian kernel.

The pre-processed functional images were then submitted to a first level statistical analysis: in each subject, a general linear model was built in which a hemodynamic response function and its time derivative were convolved with block onsets for each category and the six motion parameters entered as regressors of non-interest.

## Data-driven Analyses

### Whole-brain analyses

We implemented a mixed-model analysis of variance (ANOVA) with language (French vs. Chinese) and reading skill (typical vs. poor readers) as between-subject factors, and Category (Words vs. Faces vs. Houses) as a within-subject factor. We recovered category-specific activations through the contrasts [category X > mean of the other two categories] across the whole group (N = 96). We report effects at a threshold of p<0.001 at the voxel-level and p<0.05 family wise error (FWE) corrected for multiple comparisons at the cluster-level (denoted $p_{FWE\_corr}$).

To deepen our analyses of the effect of reading performances, we studied the correlation between reading performance (standard scores in reading test) and the word and face activation in the 96 children. We report effects at a threshold of p<0.001 at the voxel-level and p<0.05 FWE corrected for multiple comparisons at the cluster-level.

For these analyses, we displayed in the corresponding figures the results in each of the four groups (N = 24) or in each language (N = 48) at a threshold of p<0.001 at the voxel-level, non-corrected at the cluster-level to provide the reader with the full information on the activation patterns in each group.

### Mask-restricted analyses

To maximize the sensitivity to differences between groups, we focused our analyses on a restricted mask of voxels corresponding to the word-specific activation across all children determined by the [Words > Faces + Houses] contrast (p<0.001 voxel-level and $p_{FWE}$ < 0.05 cluster-level, size ~ 7497 mm$^3$). We performed an ANOVA with language (French vs. Chinese) and reading skill (typical vs. poor readers) as between-subject factors. To provide readers with full information, we report these results at the threshold of voxel-level p<0.001, non-corrected at the cluster-level.

## Literature-driven Analyses

As the number of comparisons at the voxel-level might decrease the sensitivity to small differences between typical and poor readers, we focused on Regions of Interest (ROI), which have been repeatedly shown in the literature to show a reduced activation in dyslexics. We first searched published meta-analyses of imaging studies reporting brain regions showing functional dysfunction in dyslexics in alphabetic languages (*Linkersdörfer et al., 2012*; *Maisog et al., 2008*; *Richlan et al., 2009*; *Richlan et al., 2011*). To create representative ROIs for these dyslexia-related regions, we collected all of the foci reported in each meta-analysis corresponding to the anatomical location under consideration (see *Supplementary file 4*), and averaged the reported coordinates (x, y, z respectively) to create a 6-mm-radius sphere of the averaged locus as a ROI (see *Figure 3*).

Due to the limited number of published neuroimaging studies of Chinese dyslexia, no meta-analysis was available to summarize the available evidence into a pooled estimate. However, atypical activation in a lateral prefrontal region within BA nine has been reported in Chinese children with reading disability (*Siok et al., 2004*; *Siok et al., 2009*) and this region was repeatedly found to be more involved in reading Chinese than alphabetic languages (*Bolger et al., 2005*; *Nakamura et al., 2012*; *Tan et al., 2005a*). Besides, previous studies also often reported that Chinese reading networks are more symmetrical in the ventral visual system. We thus included the foci in both left middle frontal gyrus and right occipital cortex that were reported in several meta-analyses on Chinese typical reading (*Bolger et al., 2005*; *Tan et al., 2005a*; *Wu et al., 2012*; *Zhu et al., 2014*) and created representative ROIs as above (see *Supplementary file 5*). In total, we obtained 10 ROIs related to dyslexia in alphabetic languages and three additional ROIs potentially related to Chinese typical reading and dyslexia (*Figure 3*).

We extracted the mean contrast-weighted beta values for the words vs. fixation contrast in each ROI in each child and first considered whether these values were correlated with the reading score across all participants (*Figure 3*), second entered these values in an ANOVA with language (Chinese vs. French) and reading skills (typical vs. poor readers) as between-subject factors. The false discovery rate (FDR) multiple-comparison method was implemented to take into account the multiple ROIs. We did the same analyses in the same ROIs for the contrasts of faces vs. fixation and houses vs. fixation. The FDR-corrected p value is denoted as $p_{FDR\_corr}$.

Because some of our analyses evaluated the null hypothesis of no difference between two factors, we also performed Bayesian analyses on the same data, with the same factors as the ANOVA above, using the JASP software (https://jasp-stats.org). These analyses yield a Bayes Factor (denoted BF10), which estimates the likelihood ratio of the positive (1) over the null (0) hypothesis. A BF10 of 0.20, for instance, indicates a five-times (1/0.20) greater likelihood in favor of the null hypothesis and is equivalent to the inverse notation BF01 = 5.

## Individual analyses

### Peak analyses

We further investigated whether children poor with reading might have a greater inter-individual variability in brain localization, which could putatively explain a lower activation at the group level in each voxel. We focused on those regions showing significant main effects of reading ability or language in the group activation analysis (i.e. left FFG, MFG, precentral, STS, pSTG, and SPL) and searched for active voxels (Words > fixation) in each participant in a sphere (radius = 12 mm) centered on the whole group peak coordinates in the [Words > Faces + Houses] contrast. We eliminated voxels with t-value inferior to one, and then selected the 10 strongest activated voxels within the search area. We first derived the individual center of mass of these voxels by averaging their x, y, z coordinates and calculated the distance between this center of mass and the group peak coordinates in each child. Second, we averaged the beta values measured in these voxels to obtain the maximal activation in each child. We then separately entered distance and activation into language × reading ability frequentist and Bayesian ANOVAs to investigate whether poor readers differed in peak location and activation intensity compared to typically developing children. We performed a similar analysis on the face responses in the bilateral fusiform face areas (FFA) to investigate whether poor readers had greater inter-individual variability in the location and intensity of face activations.

### Multivariate Pattern Analysis (MVPA)

Our final investigation of putative differences related to language or to reading ability was based on MVPA analysis. We focused on the regions showing significant main effects of reading ability or language in the univariate activation analysis and drew a 9 mm radius sphere centered on the averaged coordinates of foci reported in meta-analyses, and then intersected each sphere with the mask [Words > Faces + Houses] described above to obtain a fair representation of the group activations in the mask. All the voxels within the mask were included for MVPA analysis.

Secondly, within each ROI, we quantified the within-subject reproducibility of the patterns of activation by calculating in each subject the correlation coefficients between the pattern of response evoked by words relative to fixation during the first run and the pattern of response evoked by each category (words, faces, and houses) relative to fixation during the second run. The correlation coefficients were converted into Z-scores and entered into a separate ANOVA for each ROI with language (Chinese vs. French), reading ability (typical vs. poor readers), and condition (within-category correlation (i.e. words with words), vs. between-category correlation (i.e. words with faces, words with houses, faces with houses)) as factors.

We performed a similar analysis in the bilateral face fusiform areas to investigate whether poor readers showed reproducible activation patterns to faces. Bilateral face ROIs were spheres with a 9 mm radius centered on the reported peak coordinates in the face-selective activation in previous studies (left [−39, −45, −18], right [39, −45, −18]) (*Downing et al., 2006*). We intersected each sphere with the whole group activations [Face > Words + Houses] to obtain a mask (~1318 mm$^3$). All the voxels within the mask were included for MVPA analysis. We then calculated the correlation coefficients between the pattern of response evoked by faces relative to fixation during the first run and the pattern of response evoked by each category (faces, houses, and words) relative to fixation during the second run in each subject.

Note that six French typical readers and three French poor readers from *Altarelli et al., 2013* completed only one run of the visual task, so that they were not included in this MVPA analysis. For those children who had four runs, we used their first two runs to calculate the correlation coefficients between runs. The FDR multiple-comparison method was again used as a correction for the multiple ROIs tested.

## Anterior-to-Posterior organization of the ventral temporal cortex

In the following analyses, we focused on the ventral occipito-temporal region, because it is the site of one of the major changes related to reading: the emergence of a specific response to words in literates and also the site of the most reproducible hypoactivation in dyslexics compared to typical readers in alphabetic writing (*Richlan et al., 2011*). Because activations to words, faces, and houses display a gradient both along the anterior-to-posterior (i.e. 'y') axis and lateral-to-medial (i.e. 'x') axis (*Baker et al., 2007*; *Scherf et al., 2007*), we averaged the activity in successive spheres along the anterior-posterior axis at the 'x' privileged position for each visual category and compared typical and poor readers in Chinese and French participants. The spheres had a 6-mm-radius and were regularly spaced with the center positioned at y = −73, −64, −55, −46, −37, −28 respectively, x and z positions being kept constant (z = −16). The x position was based on the peak of the category-specific activation (category > others) in all participants, i.e. for Words: x = ±48; Faces: x = ±39; and Houses: x = ±30 (see *Table 2*).

We performed a separate ANOVA for each visual category with language and reading ability as between-subject factors, ROI (6 y-axis positions) and Hemisphere (left and right) as within-subject factors. In each ANOVA, we corrected for multiple comparisons using the FDR method.

## Acknowledgements

We would like to thank all the children and their parents for their collaboration in this study. We are grateful to Mengyu Tian, Weiyi Xie, Feng Ai, Manli Zhang for their help in Chinese children's behavioral and MRI data collection. We thank Le Li for providing suggestions and help on data analysis.

## Additional information

### Funding

| Funder | Grant reference number | Author |
|---|---|---|
| National Natural Science Foundation of China | 81171016 | Xiangzhi Meng |
| Agence Nationale de la Recherche | ANR-06-NEURO-019-01 | Franck Ramus |
| Fondation Bettencourt Schueller | | Stanislas Dehaene |
| Agence Nationale de la Recherche | ANR-11-BSV4-014-01 | Franck Ramus |
| Agence Nationale de la Recherche | ANR-17-EURE-0017 | Franck Ramus |
| Agence Nationale de la Recherche | ANR-10-IDEX-0001-02 PSL | Franck Ramus |
| National Natural Science Foundation of China | 81371206 | Xiangzhi Meng |
| National Natural Science Foundation of China | 31971039 | Xiangzhi Meng |
| China Scholarship Council | 201706040117 | Xiaoxia Feng |

The funders had no role in study design, data collection and interpretation, or the decision to submit the work for publication.

### Author contributions

Xiaoxia Feng, Software, Formal analysis, Validation, Investigation, Visualization, Methodology, Writing - original draft, Project administration; Irene Altarelli, Data curation, Writing - review and editing; Karla Monzalvo, Data curation; Guosheng Ding, Conceptualization, Resources; Franck Ramus, Funding acquisition, Project administration, Writing - review and editing; Hua Shu, Resources, Supervision; Stanislas Dehaene, Conceptualization, Supervision, Writing - review and editing; Xiangzhi

Meng, Conceptualization, Resources, Funding acquisition, Project administration; Ghislaine Dehaene-Lambertz, Conceptualization, Resources, Data curation, Supervision, Funding acquisition, Project administration, Writing - review and editing

### Author ORCIDs
Xiaoxia Feng (iD) https://orcid.org/0000-0002-0414-4509
Irene Altarelli (iD) https://orcid.org/0000-0003-0895-0556
Xiangzhi Meng (iD) https://orcid.org/0000-0003-1265-2332
Ghislaine Dehaene-Lambertz (iD) https://orcid.org/0000-0003-2221-9081

### Ethics
Human subjects: The study was approved by Institutional Review Boards at Beijing Normal University in China and local ethics committee (CPP Ile de France VII, N° 11-008) in Kremlin-Bicêtre, France (#20130331). Written consent and consent to publish was obtained from all children and their parents.

### Decision letter and Author response
Decision letter https://doi.org/10.7554/eLife.54591.sa1
Author response https://doi.org/10.7554/eLife.54591.sa2

## Additional files
### Supplementary files
• Supplementary file 1. Demography and performance on literacy tests for Chinese children.

• Supplementary file 2. Demography and performance on literacy tests for French children.

• Supplementary file 3. Regions of significant activations for each visual category vs. the two others in each group (individual voxel p=0.001, cluster-level FWE corrected).

• Supplementary file 4. Summary of activation foci in meta-analyses of dyslexia in alphabetic languages.

• Supplementary file 5. Summary of foci in meta-analyses of reading in Chinese.

• Supplementary file 6. Distance between individual center of 10 most activated voxels and group peaks.

• Transparent reporting form

### Data availability
The processed data to generate figures in this manuscript will be shared in the Open Science Framework (Identifier: https://doi.org/10.17605/OSF.IO/C4AXG).

The following dataset was generated:

| Author(s) | Year | Dataset title | Dataset URL | Database and Identifier |
|---|---|---|---|---|
| Feng X | 2020 | Universal brain mechanisms of reading acquisition: an fMRI study in normal and dyslexic French and Chinese children | https://doi.org/10.17605/OSF.IO/C4AXG | Open Science Framework, 10.17605/OSF.IO/C4AXG |

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
