## [Decision Letter]

**Acceptance summary:**

Your contribution is important because it shows common neural correlates for word recognition in typical developing and dyslexic children in French and Chinese. While the conclusion is based on "null effects" (i.e., Chinese and French dyslexics are similar), a set of exhaustive contemporary approaches (whole-brain MRI, ROIs, MVPA, Bayesian sensitivity analysis) gather consistent support the findings. The results provide additional insight into the ongoing debate about the universality of the neural correlates of dyslexia across cultures and writing systems. The study fills an important gap, going beyond the existing meta-analytic literature which claimed that the neural basis of dyslexia is similar across alphabetic writing systems despite difference in orthographic depths in the visual word form area (VWFA). The current work extends this claim by showing experimentally that even in completely different writing systems such as Chinese and French, the neural correlates are shared. The authors took incredible care to match across the French and Chinese languages in terms of the various parameters (scanner characteristics and demographics).

**Decision letter after peer review:**

Thank you for submitting your article "Universal brain mechanisms of reading acquisition: an fMRI study in normal and dyslexic French and Chinese children" for consideration by *eLife*. Your article has been reviewed by three peer reviewers, and the evaluation has been overseen by a Reviewing Editor and Floris de Lange as the Senior Editor. The following individual involved in review of your submission has agreed to reveal their identity: Charles Perfetti (Reviewer #3).

The reviewers have discussed the reviews with one another and the Reviewing Editor has drafted this decision to help you prepare a revised submission.

Here I have summarised the reviews to help your revision. I intend that you will use my letter as a guide as to what you should address in your revision, and refer from my points in the editorial letter to the 3 reviews as reference, such that you can read about the points I mention in the letter in more depth in the reviews. I will mainly draw your attention to the concerns from reviewers 1, 2, and 3 regarding the interpretations of your data, and the strength of the support that your statistical analyses offer to those conclusions.

The editorial concerns below must be addressed in any revision. I include the full reviews because they are highly constructive and contain feedback that can help you improve the manuscript; I encourage you to incorporate as much of the reviewer's feedback as possible. Below I will summarise the changes necessary for the revision:

1) Bayesian Sensitivity Analysis – Like Reviewer 1, I am particularly worried about how weak the Bayes factor evidence is; at the moment it is weak and the prior that was chosen is not very informed or reflective of either direction of assumptions. Thus, a sensitivity analysis where a variety of priors, representing different assumptions, should be carried out, including an informed prior based on previous literature and which stems from the contrasts suggested by Reviewers 2 and 3. For more about sensitivity analysis, please see Oakley JE, O'Hagan A. Probabilistic sensitivity analysis of complex models: a Bayesian approach. Journal of the Royal Statistical Society: Series B (Statistical Methodology). 2004 Aug;66(3):751-69.

2) Whole Brain Analysis – From Reviewer 2, I found it important to address their point that "the "universality of dyslexia" argument (assumes) there is a neural signature of dyslexia. Yet, at the whole brain level, there is no significant difference between dyslexia and controls. The rest of the analyses heavily relies on the regions identified in the ROI based approach. This should be thoroughly discussed and addressed as an important limitation."

3) By Language Analysis – Both Reviewer 2 and 3 would like to see analyses broken down by writing system. From Reviewer 2 "Since authors do show significant correlation with reading abilities in several areas, there appears to be room to explore differences in writing systems. Can the authors do further analyses and potentially build on these results (also), and if not, justify why?"

4) Discussion of Power – All reviewers were concerned about the experiment and contrasts being properly powered. Though N=96, there were only 24 per group for each of the four groups. As Reviewer 2 asks, on what basis is this sufficiently powered? On a related point, given the specificity of the claims wrt a lack of language interaction, Reviewer 3 points out that planned comparisons focused on language (if power is sufficient), in both ANOVAs and Bayes factors, are needed to support this line of argumentation.

5) Discussion of whether and how the population was identified, and reading skill instead of dyslexia – Reviewer 3 had a very helpful and constructive point: "the value of the paper is adding incremental evidence for a shared pattern of activation areas that distinguish low levels of reading skill from more typical, higher levels." But thinks that framing this result in terms of *reading skill* is more warranted than in terms of dyslexia, given the way Chinese dyslexics were identified from the -1.25SD of normal classrooms.

6) Interpretation of the results in relation to existing literature and LMFG – Lastly, the paper's main conclusion is that these results counter existing research showing brain activation patterns implicating the involvement of LMFG in logographic more than alphabetic reading. In fact, as Reviewer 3 points out, the results reported here actually align with existing studies reporting differences between logographic and alphabetic systems, and this should be highlighted. Or to put it another way, now that the LMFG is found to be involved in alphabetic reading, is it now elevated to part of the universal reading network that has previously excluded it?

Please find the full reviews attached for your reference:

Reviewer #1:

1) Reasoning stems from a fallacy – namely that the lack of evidence for a difference is evidence for a lack of difference. It is not.

2) It is problematic that the authors interpret BOLD signal/ brain activity as somehow equivalent to a mechanism or computation, without any attempt to constrain or define that computation.

3) The study of dyslexia is important, but here it clouds the inference waters about the neural substrate of reading – it cannot separate what the neural "causes" vs the neural "effects" of dyslexia are.

4) The experiment is designed to rely on interactions for scientific inference, yet appears only to be powered to detect main effects – though no power analysis is presented. The Bayes factors reported are weak evidence for the null, and worse are highly dependent on the prior. Using a more informed prior (for example, from the existing literature cited here, such as 9,10,32 and many others) will decrease evidence for the null dramatically (and killing this Bayes factor further, I suspect; moreover, it is only Bayes factors > 10 that are regarded as strong evidence for the null). That has not been done here. Ideally one needs to show a comparable range of Bayes factors with multiple, varied, or highly justified priors. That has not been done here. Mixed models should be used here over anova in order to take into account item variance simultaneously with subject variance.

5) It is not clear how this work is an advance over existing findings, see in particular citations 9, 10, and 32.

Reviewer #2:

Summary

In the current manuscript, authors show convergent results of common neural correlates of word recognition in typical developing and dyslexic children in French and Chinese. While their conclusion is based on "null effects" (i.e., Chinese and French dyslexics are similar), they use a set of exhaustive approaches (whole-brain MRI, ROIs, MVPA) and Bayesian inferences to support their findings. The results provide additional insight into the ongoing debate about the universality of the neural correlates of dyslexia across cultures and writing systems. Their results also shed light on the neural correlates of pediatric dyslexia in children showing similar characteristics to adults (though they do not compare with adults directly in their study). The study fills an important gap that goes beyond prior meta-analytic literature that showed that the neural basis of dyslexia is similar across alphabetic writing systems that are different across orthographic depths in the visual word form area (VWFA); they show using empirical data (and not meta-analyses) that even completely different writing systems such as Chinese and French are similar. The authors make pain-staking and impressive efforts to match across French and Chinese in terms of the various parameters (scanner characteristics and demographic). The results are generally convincing and manuscript well written. Detailed major comments are listed below.

1) Premise of the study: The major issue for this reviewer was that the "universality of dyslexia" argument lies on the fact that there is some neural signature of dyslexia. Yet, at the whole brain level, there is no significant difference between dyslexia and controls. The rest of the analyses heavily relies on the regions identified in the ROI based approach. This should be thoroughly discussed and addressed as an important limitation.

2) Correlation analyses with reading: Authors then move into correlation with reading ability but do not pursue further than showing correlations in the entire sample. They do not break the sample up into writing systems. Since authors do show significant correlation with reading abilities in several areas, there appears to be room to explore differences in writing systems. Can the authors do further analyses and potentially build on these results (also), and if not, justify why?

3) Power & sample size: Discussion paragraph two: Authors mention that they were sufficiently powered because they had N=96. This is still 24 per group for each of the four groups. On what basis was it sufficiently powered? Especially in reporting these "null" results, it's important to provide evidence and perhaps reiterate this evidence when mentioning their "sufficient power" in the Discussion.

4) MVPA: In general MVPA was confusing in many ways and the reviewer recommends a major rewrite.

a) Motivation: Introduction paragraph seven: The phrase about MVPA where they motivate this analysis should probably be a separate sentence as the question being asked is very different than the first part of the question. Further, this justification comes out of nowhere and should be elaborated. The authors claim that they wish to test dispersion and intensity using classical analyses, and also using MVPA. They also do not mention in the Introduction, but in the Materials and methods, mention that they are examining stability (reproducibility from Run 1 to Run 2). Please clarify in the Introduction.

b) Materials and methods subsection “Multivariate Pattern Analysis (MVPA”: This was very confusing all around. For example… (1) What values are fed into as features? Time-series correlation? Or betas for each trial? They mention "pattern of response evoked by words relative to fixation" but this is unclear. (2) They claim they are doing multi-"voxel" pattern analyses (MVPA), but they also seem to refer to each ROI having one correlation coefficient. (3) It was not clear why this is called MVPA. What was the analytical technique used, ANOVA? Correlation-based MVPA (what is this exactly?)? There are probably many more, but it's hard to judge the quality of this analyses based on their lack of details and hard to ask further questions at this point.

5) Population: The test scores in Chinese was normed based on their own data of 2554 individuals. It is not clear whether this is appropriate (though size is large). How were these children recruited; is this a nationally representative population? In addition, how French children with dyslexia was identified (and criteria) was not mentioned in this paper and the authors simply refer to another paper. Since this is quite important, it will be nice to have this detailed here in this manuscript as well.

6) Reading ability: (1) How did they combine reading skills across writing systems to combine all in a correlation analysis, especially when reading measures that fed into the dyslexia criteria were different, there is no standardized measures in Chinese and just that the measures are so different across the board? (2) This also means that although unreported, the authors can compare reading scores (as well as others such as IQ) across writing systems, and report in Table 1? Please elaborate.

Reviewer #3:

General Assessment. An interesting paper with rich results, an fMRI study comparing previously reported results of French children with new data on Chinse children. The study is characterized by careful sample matching, comparable instrumentation across sites, and multiple sophisticated analyses. The paper provides a context of previous brain imaging studies that compare dyslexia across Chinese and alphabetic writing and places its results in this context. Its message is not novel-that the brain shows a reading network that is shared across languages and writing systems-but because the research has suggested different conclusions on whether reading Chinese and reading alphabetic writing involve different brain areas, these results could add weight to two aspects of the issue. Do alphabetic and Chinese reading engage the same brain areas to the same extent? Do Chinese and alphabetic dyslexics show similar activation patterns across brain areas? The results answer the first question by "yes" and no", but the authors emphasize more the "yes" part. They answer the second part "Yes" Some revision and perhaps reframing of the contribution of the paper to the two issues engaged-language/writing system factors and dyslexia-would improve the paper.

Major comments on context and Interpretation.

1) One suggestion is to broaden the context for the question and to refine its form. Arguments that the procedures of reading are universal have been made on the basis of behavioral studies and the logic of writing systems (e.g. Perfetti, 2003; other observations by Mattingly, 1972) If graphs-the written symbols of language-encode linguistic forms-then the brain, to decode writing, must connect visual areas to language areas. Beyond that fundamental universal, the factors of mapping of phonology and morphology, and perhaps graphic complexity, could influence the details. Whether there is a shared reading circuit across languages has not been the main question. Rather, the question has been, given a shared or universal circuit, what differences might one expect based on the features of the writing system and the language?

2) Although the authors' argument is strong on the universalist side, the data show convergence with the results of Siok, Tan et al. in the greater LFMG and SPL activation for Chinese. Puzzling, as the authors note, is the difference in pSTG, also higher for Chinese, contrary to other results. Given these language differences, what does the failure to find a language x dyslexia interaction mean? Both normal and dyslexics in Chinese show the Chinese pattern (more activation in MFG and SPL) and both controls and dyslexics in French show the French pattern (Less MFG, Less SPL, and, surprisingly, pSTG), as shown in Figure 4.

3) With these language differences, an appropriate interpretative framing would be to emphasize writing system/language differences while finding no dyslexia differences across languages. The paper reports and comments on the differences findings but does not highlight them. The Abstract and conclusion should be written to reflect these findings.

4) The LFMG difference might reflect the role of writing experience in Chinese, which may make frontal brain areas near the motor cortex, especially Exner's area, more active in reading. The authors cite the comparison of French and Chinese by Nakamura et al., 2012), but the evidence from Cao & Perfetti, 2017, is also relevant because it reports that the overlap of writing and reading brain areas is greater in Chinese than English and also in learners of Chinese as L2 who had learned through character writing compared with learners who had learned through pinyin.

5) More generally on the issue of context and interpretation, the authors should consider the relevance of Perfetti, Cao, and Booth (2013), which provides a more nuanced framing of the universal/specialization issue and summarizes results from children's studies carried out by Booth et al. to identify brain regions that show developmental changes in English and Chinese, highlighting commonalities along with divergence in STG and LMFG.

6) Any comments on why MFG shows a much stronger dyslexia effect for French (strong) than Chinese children (marginal or null), according to the Bayes factor data shown in Table 4? The language difference shows more MFG involvement among Chinese than French. The reporting of the results in subsection “ROI-Level analysis” refers only to the Bayesian evidence against a language x dyslexia interaction, ignoring the post-hoc analyses (which given the goal of the study, could have been planned comparisons), which also show strong F10 support for an MFG difference between French and Chinese dyslexia but marginally "moderate" support for such a difference between French and Chinese controls.

7) Concerning the MFG, a fair summary of the results is that MFG activation was observed more for Chinese than French and more for controls than dyslexics. It would be appropriate to point out that this confirms conclusions from other research. Thus, the conclusions from Siok, Tan, et al. that the LMFG is more involved in reading Chinese than in alphabetic reading is also found here, as is their conclusion of its under-activation in Chinese dyslexia. What may be new in the present paper is evidence that under-activation of the MFG may also characterize reading French. This last conclusion is not based on the present results, however, but on results from the previously reported results for the French children. The authors should point out whether the original publication of the French results reached this conclusion or whether it emerged only in the comparisons of the present paper. The Monzalvo et al. abstract emphasizes VWFA factors in dyslexia, but not MFG.

8) The Interpretation of the VWFA as a universal site should take into account that both faces and houses showed lower activation for dyslexics along the anterior-posterior axis. This affects how to interpret the lower activation for dyslexics to words in the VWFA.

9) The conclusion of the paper needs to be revised to more accurately reflect the results of the study. This should include the differences that were found, not just those that were not found and a more contextualized grounding for the conclusion that "cultural variability is merely reflected in the variable emphasis that different writing system put on (linguistic units)". This is not new (which is OK; it's worth emphasizing) and has been pointed in other papers in which the results have shown language differences.

10) The authors acknowledge a possible task issue but an additional consideration seems relevant. The task involved reading only incidentally. We know that for a skilled reader, implicit reading engages the network non-attentively (automatically), so the task itself is ok, as are naming and semantic judgement used in other studies. The problem is that, because it is the only task used, we get a limited picture of the reading network, perhaps especially for the dyslexics who cannot be assumed to engage the reading network automatically. It is worth noting that the research reporting Chinese-alphabetic differences did not use passive reading tasks but naming, rhyming, and meaning decision tasks. I don't know whether this is relevant for the differences, but It would strengthen the contribution of the paper if it could suggest a reason for different conclusions arising in the literature or address the task issue in that context.

11) I also wonder about the degree of reading difficulties shown in the sample. The Chinese children were classified as dyslexic by showing a -1.25 SD score on the CCRT character recognition test. We need to know what the task was for the CCRT. Performance on a recognition task reflects learning through experience. A reading problem may reflect a poor learning outcome from experience but should not reflect mere experience. Also -1.25 SD seems to identify a rather large percentage of children as dyslexic, compared with other studies, which may make it more likely that reading experience is involved to some extent. This is not a problem for an approach that treats reading as a graded skill, which the authors do in some of their analyses. Making reading skill rather than dyslexia the framing of the paper could help with this.

12) The ANOVA reported in subsection “Data-driven Analyses” does not provide a main effect of category. Instead it tests one category against the mean of the other two. And this leads to multiple tests that are appropriate with a significant main effect. A standard ANOVA with 3 levels of the category variable is needed.

13) Another statistical issue is the use of separate planned comparisons within each language when no interaction involving language is reported. In subsection “Anterior-to-Posterior gradient in the visual cortex”, this comparison, without a report of a significant language interaction, is shown for the anterior-posterior gradient analysis allowing the suggestion that the only dyslexia difference for both languages was near the VFWA. In other analyses, the authors report a lack of a language interaction and no further within-language comparisons. Given the focus of the research on language differences, planned comparisons between languages is quite reasonable to do in all analysis but the authors seem to have done this only selectively. Some explanation is needed.

14) The sentences "Dyslexics might have more dispersed activations without focal peaks that would also create weaker responses at the group-level, but a reproducible pattern of activations. In that case, the within-subject reproducibility of multivariate activation patterns should not differ between normal-readers and dyslexics" from subsection “MVPA of the activations to words”. The logic here escapes me. If Ds have more dispersed activation without focal peaks their reproducibility of multivariate patterns could still differ from controls. More voxels but not the same voxels, reflecting intrinsic variability. I think the authors are right that the individual peaks analysis means the between-group variability is not due to averaging individuals. But the differences in these individual MVPA analyses mean that variability is higher with dyslexics. The reason for this should be considered.

15) Why on the MVPA are the data analyzed transformations of correlations of first run with second run? The idea is reproducibility, but the second run is a second run that provides a familiarization component that may affect controls and dyslexics differently.

16) Another point to be explained is why some analyses use fixation as the comparison (MVPA) whereas most use the contrast of one condition vs the other two.

17) Figure 2 shows areas with correlations with reading scores. Either the caption or the images should make clear the directions of the correlations and make clear this is for all subjects. Given the goal of the study, showing these separately for French and Chinese is better, even though it reduces the power.

18) Figure 3A is a bit puzzling. It shows peaks from literature on Chinese and alphabetic studies, but the only peak in LH is at MFG for Chinese. Certainly, studies of Chinese find peaks in other LH regions. Do the peaks have the same definition across studies and meta-analyses? One peak for each region of interest or region of a hypothesized reading network probably would show a different picture. As it stands the visual here contradicts universality and may be misleading. As for 3B, again it would be useful to indicate the direction of the correlations or state they are all positive if that is the case.

19) The authors say that the LFMG dysfunction "was previously claimed to be specific to Chinese dyslexia". This is more accurately described as a discovery of an LMFG deficit in Chinese dyslexics in the absence of any existing evidence of a comparable deficit having been reported in in studies of alphabetic languages. The previously published study of French children should have added the discovery that LFMG dysfunction can characterize alphabetic reading as well.

[Editors' note: further revisions were suggested prior to acceptance, as described below.]

Thank you for resubmitting your article "A universal reading network and its modulation by script type and reading abilities in French and Chinese children" for consideration by *eLife*. Your revised article has been reviewed by one peer reviewer, and the evaluation has been overseen by Andrea Martin as Reviewing Editor and Floris de Lange as the Senior Editor. The following individual involved in review of your submission has agreed to reveal their identity: Charles Perfetti (Reviewer #3).

The reviewers have discussed the reviews with one another and the Reviewing Editor has drafted this decision to help you prepare a revised submission.

Summary:

The reviewer and myself were encouraged by the Bayesian Sensitivity Analysis performed. However, the reviewer has 4 concise but excellent suggestions that I am confident you can address.

Please address points 1-4 by the reviewer; each query is concrete, concise, addressable, and clearly laid out. Please make these revisions and clearly indicated in the revised manuscript text as well as describe in a response letter how you have handled each point.

Reviewer #3:

The authors made several significant improvements in the paper and were responsive to reviewers' comments. I think the strength of the paper lies in the developmental results for a Chinese sample to compare with previously published results of a French sample, along with a better portrayal of its contribution in the context of other research. The French-Chinese comparison itself remains a question for me, but the Abstract, Discussion, and conclusion are more in line with the results.

I suggest the authors consider a few additional revisions.

1) The authors' reframing from dyslexia to reading skill is important, given the likely differences in reading skill between the Chinese and French samples. But the use of the word "impaired" to refer to lower skill readers throughout the ms does not follow through on this reframing. Dyslexia is an impairment. It might be that the French sample includes mainly or exclusively impaired readers and the authors may prefer that term to be consistent with their original publication of the French data. If so, it might be enough to have a sentence or two that explicitly address the terminology shifts or point out the possible French-Chinese differences in the low skill group. If effective reading experience is what is involved especially in Chinese-i.e. the rate at which a child has benefitted from learning opportunities, including reading practice-this might be important to mention. This would be consistent with the demands of incremental character learning and practice. It may be interesting that the distinction between impairment and delay may not matter given the authors' interpretation of the results of their comparisons.

2) If I understand correctly, the important conclusion that LMFG is part of the universal network is indirect, dependent on adding a Chinese sample to the original French sample for comparison. The French result did not show significant under-activation for impaired readers. Instead, only when the Chinese sample was included did this occur. I think this point should be made explicit. Maybe I misunderstand something, but it seems that only by adding a Chinese sample that we see LMFG problem in French children with reading problems.

3) The greater activation for Chinese than French in the posterior superior temporal gyrus/sulcus continues to be a bit of puzzle. The explanation on 19-20 does not seem very plausible. Children's use of PinYin might help explain why the PSTG in Chinese is not less activated than in French, contrary to expectations based on decoding accounts of the PSTG. However, I don't see how it can explain the reversal that was observed for children who have been reading characters for 3-5 years. Rather than this extended speculation, it might be better to just mention it as one of several possibilities, including the possibility of more complex syllable decoding because of Chinese tones and connections to spoken language functions, or a need to suppress the activation of the syllable associated with the character (because it leads to competition for character identification), or others that also are speculative. If the evidence for a letter-phoneme function of this area comes mainly from explicit reading tasks, maybe this area is doing something else related to the target task during incidental reading.

4) The authors use "script" where they actually refer to writing system. I strongly suggest a change in the title and elsewhere to writing system. The difference between French and Chinese is one of writing system and this inevitably brings along a difference in script. The difference between French Cursive and French Gothic and Chinese Traditional and Chinese Simplified are differences of script.

---

## [Author Response]

The editorial concerns below must be addressed in any revision. I include the full reviews because they are highly constructive and contain feedback that can help you improve the manuscript; I encourage you to incorporate as much of the reviewer's feedback as possible. Below I will summarise the changes necessary for the revision:1) Bayesian Sensitivity Analysis – Like Reviewer 1, I am particularly worried about how weak the Bayes factor evidence is; at the moment it is weak and the prior that was chosen is not very informed or reflective of either direction of assumptions. Thus, a sensitivity analysis where a variety of priors, representing different assumptions, should be carried out, including an informed prior based on previous literature and which stems from the contrasts suggested by Reviewers 2 and 3. For more about sensitivity analysis, please see Oakley JE, O'Hagan A. Probabilistic sensitivity analysis of complex models: a Bayesian approach. Journal of the Royal Statistical Society: Series B (Statistical Methodology). 2004 Aug;66(3):751-69.

Following editor and the reviewers’ suggestions, we have conducted sensitivity analyses in the post-hoc pairwise comparisons (as supplementary material, see below). We now report the results in each language as well as between languages (please see main text and figures below).

“We used Bayesian ANOVAs on the Word activation to assess the likelihood of the null hypothesis H0 over H1 in the case of the interaction Language × Reading skills. As also explained in the Materials and methods, the Bayes factor (BF10) is the ratio of the amount of evidence for H1 above H0 (BF01 for H0 above H1). Evidence for H1 against H0 is generally considered as moderate for BF10 ≥ 3, and strong for BF10 ≥ 10. Bayesian analyses provided similar conclusions to frequentist analyses for the main effect of Language and Reading skills (Table 4). The BF10 for an effect of language was 80.21 in the MFG, 7.05 in the SPL and 9.91 in the pSTG. The likelihood of an effect of Reading skill in the FFG was BF10 = 1528.97 higher than that of the null hypothesis; the same BF10 was 37.51 in the STS, 9.95 in the MFG and 5.63 in the PCG respectively. It was also the case in the post-hoc analyses within each language (Table 4). Notably, there was strong evidence of a reading skills effect in the fusiform gyrus (FFG) and moderate evidence in the STS and PCG in both languages. In the pSTG and also paradoxically in the MFG described as a dyslexic marker in Chinese, the evidence of a reading skill effect was strong in French (BF10 = 23.74 and 29.35 for each site respectively) but absent or weak in Chinese children (BF10 < 1 and 1.62). Sensitivity analysis revealed that the Bayes factor stayed about the same for a wide range of prior specifications (Cauchy prior width: 01.5) in the FFG, STS, PCG and MFG for the comparisons between controls and impaired readers in each language except for the pSTG in Chinese children (see Author response image 1)”.

Turning now to the effect of language in each reading group, differences between languages were mainly observed between the groups of impaired readers: Chinese children showed larger activations than French in the MFG (BF10 = 309.65) and pSTG (BF10 = 74.01) although the same tendency was present in controls. The above Bayes factor stayed the same for a wide range of prior specifications (Cauchy prior width: 0-1.5) in the sensitivity analysis (see Author response image 2). However, there was no evidence for a significant interaction Language × Reading skill in all these ROIs. On the contrary, the null effect was supported by moderate evidence in the FFG, MFG, STS, PCG, and SPL (respectively BF10 = 0.31, 0.31, 0.31, 0.27 and 0.32, i.e. the likelihood of the null hypothesis BF01 = 3.23, 3.23, 3.23, 3.70 and 3.13); there was no evidence in either direction in pSTG (BF10 = 1.05) (Table 4).

**Author response image 1. sa2fig1:** Sensitivity analysis with a variety of priors in the post-hoc pairwise comparisons between controls and impaired-readers in each language (Left: Chinese children; Right: French children).

**Author response image 2. sa2fig2:** Sensitivity analysis with a variety of priors in the post-hoc pairwise comparisons between Chinese and French children separately in control and impaired-reading children (Left: Controls; Right: impaired-readers).

We would also like to point out that, contrary to the reviewer’s worries, we have strong positive evidence, using both Bayesian statistics and frequency statistics, in the main ROIs described in the literature as sensitive to reading performance or proposed to be Chinese-specific. In particular, 2 sites, the FFG and the MFG which have been hypothesized to be crucial for reading acquisition respectively in alphabetic (FFG) and Chinese (MFG) languages are significantly activated and showed significant differences between impaired and typical children in each language. Activation in the VWFA has been emphasized as the main marker of reading skills in alphabetical languages but there is disagreement on its role in other writing systems, notably Chinese because of the large differences in visual shapes, phonological content and relation with handwriting between Roman letters and Chinese characters.

The likelihood of an effect of Reading skill in the VWFA region was strong in both French and Chinese children: BF10 = 93.16/13.58 in agreement with Krafnik et al’s (2016) conclusion of script invariance in that region in first graders. Likewise, the MFG (which has been described as the main marker of reading difficulties in Chinese, but not in alphabetic languages), exhibits a strong effect in French children but a weak one in Chinese: BF10 = 29.35/1.62, i.e. the converse of what was predicted by the culture-specific model, thus emphasizing a more universal role of this region.

Finally, the likelihood of a significant interaction between language and reading ability as suggested by the past literature on these regions, was not at all supported. It resulted in a BF10=0.31 in both cases (thus a moderate likelihood for the *absence* of interaction).

In summary, our results provide positive (and relatively strong) evidence that the VWFA and the MFG are involved in both alphabetic and logographic writing systems even if, indeed, there is modulation of the activation in some of these areas by different factors including the writing system and reading competence, a point which we now discuss in more detail.

2) Whole Brain Analysis – From Reviewer 2, I found it important to address their point that "the "universality of dyslexia" argument (assumes) there is a neural signature of dyslexia. Yet, at the whole brain level, there is no significant difference between dyslexia and controls. The rest of the analyses heavily relies on the regions identified in the ROI based approach. This should be thoroughly discussed and addressed as an important limitation."

Although our groups were larger than in any previous study comparing Chinese and alphabetic writing systems, and were also more homogeneous in age, whole-brain analyses at corrected-level of significance failed to show significant differences between reading-impaired children and typical children. However, the most important results in our study do not arise from a binary group comparison but from the correlation of reading scores with brain activity. When we analyzed a more continuous variable (the reading scores), we recovered significant correlations in classical regions of the reading circuit in Chinese and French participants at the whole brain level (As suggested by reviewer 3, we now present results in each group).

In many brain imaging studies in children, comparisons between groups are often not significant at the whole brain level (see for example Pleisch et al., (2019) testing 18 vs 20 first-grade German children with poor and typical reading skills, and at varying familial risk for developmental dyslexia). This is probably due to the higher variance between children within each of the groups in a variety of factors: brain maturation, learning experience, attention and focus in the task, etc. All of these factors are different in adults who can remain attentive, perform more consistently, and are generally at the peak of the skill being tested (i.e., no longer learning). Furthermore, the difference between typical readers and dyslexics may be smaller in children than in adults because even the normal children are still in the process of learning and are therefore not fully competent, whereas the adult typical readers (often recruited at university) are at ceiling.

The small effect size in our study may be partly due to the definition of our groups, which included reading-impaired children that were not full-blown dyslexics. Indeed, the reading delay in Chinese children was less than in French children (Table 1), but even in French children who were typical French dyslexics (more than 2 years of delay in the LUM test), whole-brain analyses were poorly sensitive, which underlines the difficulties of pediatric research. Finally, as has been shown in alphabetic languages, the slope of normal reading acquisition in children depends on the transparency of the writing system. In other words, a typical 10-year-old reader would not perform equally well in English, French, German or Italian (even if we could use exactly the same test) because he or she has not yet attained the same level of reading fluency. Thus, comparing languages and assessing equivalent reading delay in different writing systems, while still matching for age and education, raises complex issues and increases the difficulties of intercultural comparisons in children compared to adults.

However, since we had a particular interest in understanding the role of previously identified brain regions, an analysis based on ROIs was a reasonable choice, having the great advantage of mitigating the problem of multiple comparisons and reducing the risk of false negatives, but also allowing us to examine activation patterns at an individual level, as in the MPVA analyses. These targeted analyses were able to show the role of MFG in alphabetic languages. As usual in science, the results will need to be confirmed in future studies using larger samples to obtain more robust inferences.

We have added above points at the end of the discussion as limitations as suggested.

3) By Language Analysis – Both Reviewer 2 and 3 would like to see analyses broken down by writing system. From Reviewer 2 "Since authors do show significant correlation with reading abilities in several areas, there appears to be room to explore differences in writing systems. Can the authors do further analyses and potentially build on these results (also), and if not, justify why?"

We have added the requested analyses and performed the correlation analysis separately in each writing system. As predicted by Reviewer 3, the split decreased the statistical power. To be fully informative for the reader, we presented results at the threshold of voxel-level p<0.001 and cluster-level uncorrected (Figure 2B and 2C). As it can be seen, similar regions (left STS, bilateral PCG, bilateral MFG, bilateral FFG, right MOG) were sensitive to the reading performance in the two languages.

4) Discussion of Power – All reviewers were concerned about the experiment and contrasts being properly powered. Though N=96, there were only 24 per group for each of the four groups. As Reviewer 2 asks, on what basis is this sufficiently powered? On a related point, given the specificity of the claims wrt a lack of language interaction, Reviewer 3 points out that planned comparisons focused on language (if power is sufficient), in both ANOVAs and Bayes factors, are needed to support this line of argumentation.

The issue of power in fMRI studies is an extremely complicated one, very different from behavioral studies with a single dependent measure. There is no agreed-upon way to precalculate power for an fMRI study, which involves typically ~200,000 voxels, each with their own level of signal and noise.

Furthermore, power cannot be computed a-posteriori. Let us cite from Mumford’s (2012) reference article on this topic: “Due to the relationship between the Type I error rate and power, running a power calculation on a dataset to predict the power of that study is not informative (Hoenig and Heisey, 2001; Levine and Ensom, 2001). […] What can you do if you do not detect anything significant for one of your hypotheses and reviewers request a posthoc power analysis? Since it is not informative and there are plenty of references explaining why (Goodman and Berlin, 1994; Hoenig and Heisey, 2001; Levine and Ensom, 2001), you can respectfully decline to run a posthoc power analysis. One alternative, which is often suggested, is to supply a confidence interval for the estimate (Goodman and Berlin, 1994). The reason behind this is that the confidence interval provides intuition for the range of values supported by the data that we do have.”

Besides, it is generally recommended to have at least 20 subjects in fMRI activation studies to obtain stable results (Mumford, 2012). In the meta-analysis on reading presented by Martin et al., 2015, the average number of children (7-12 years) in 20 studies was 19.75 (5 to 64) compared to 14 in the adult studies (4 to 37) and for the meta-analysis on dyslexia (Richlan, et al., 2009), the average number of participants was 23 in each group (12 to 41). The fMRI studies comparing writing systems in children also had a similar number of participants: 8 children in Siok et al., 2004 and 12 adolescents in Siok et al., 2008, 17 first graders in Krafnick et al., 2016,

21 adult participants per language (Spanish, English, Hebrew, and Chinese) in Rueckl et al., 2015, and 34 simultaneous Hindi-English 9 year-old readers in Cherodath et al., 2015. Thus, even considering each subgroup, with N=24, we are in a classic range of participants in fMRI studies. Furthermore, given the differences between writing systems and some previous studies and especially when we focused on specific ROIs, we should have expected large interactions between language and reading system. Not only did we not find such large interactions, but we found moderate Bayesian evidence in favor of the null hypothesis of no interaction, and positive evidence that the effects were present in each writing system (see preceding point).

We added a discussion of those points in the main text, and well as the post-hoc comparisons in each language.

5) Discussion of whether and how the population was identified, and reading skill instead of dyslexia – Reviewer 3 had a very helpful and constructive point: "the value of the paper is adding incremental evidence for a shared pattern of activation areas that distinguish low levels of reading skill from more typical, higher levels." But thinks that framing this result in terms of *reading skill* is more warranted than in terms of dyslexia, given the way Chinese dyslexics were identified from the -1.25SD of normal classrooms.

We thank the reviewer for this suggestion, with which we fully agree. We have replaced “dyslexia” by “impaired readers” and “reading-impaired children”. In the Results section, we used “the effect of reading skill” rather than “the effect of dyslexia” to describe the corresponding results. We also revised the Discussion section.

6) Interpretation of the results in relation to existing literature and LMFG – Lastly, the paper's main conclusion is that these results counter existing research showing brain activation patterns implicating the involvement of LMFG in logographic more than alphabetic reading. In fact, as Reviewer 3 points out, the results reported here actually align with existing studies reporting differences between logographic and alphabetic systems, and this should be highlighted. Or to put it another way, now that the LMFG is found to be involved in alphabetic reading, is it now elevated to part of the universal reading network that has previously excluded it?

We agree that we have emphasized the commonalities between writing systems, and not discussed enough the language specificities that we observed. We have corrected our discussion in this respect, and notably discussed the functional significance of the LMFG in Chinese and alphabetic languages. Consistent with our previous study (Nakamura, et al., 2012), we propose that LMFG is one of the region that participate in reading in all cultures, but that its activation intensity is modulated by the learning experience, e.g. more activation in Chinese due to more writing practice in Chinese children than children in alphabetic language.

“The greater involvement of the left middle frontal gyrus (BA9) in Chinese than in French children is in line with existing research (Bolger et al., 2005; Tan et al., 2005a). This region has been proposed to support language-specific processing required by Chinese reading, such as whole-syllable retrieval (Tan et al., 2005a), the tonal nature of Chinese phonology (Gandour et al., 2002), memory-based lexical integration of orthography, phonology and semantics (Perfetti et al., 2005), visuospatial working memory (Wu et al., 2012) or writing gesture information (Cao and Perfetti, 2016; Nakamura et al., 2012). In the reading of English, its proposed functional role is also diverse: lexical semantics (Bolger et al., 2005), phonological processing (Pugh et al., 1996), lexical selection (Bolger et al., 2008), or grapheme–phonology conversion (Jobard et al., 2003).

Perhaps the most likely explanation is that the LMFG, as well as the more medial and dorsal Exner area, are systematically involved in writing (Planton et al., 2013; Purcell et al., 2011). FMRI studies in typical adult readers, comparing French and Chinese reading, have shown that the left MTG is not specific to Chinese writing, but includes a representation of handwriting gestures that is engaged in both alphabetical and non-alphabetical languages (Nakamura et al., 2012). As proposed by Nakamura et al., 2012, the activation in the MFG and PCG, as seen here, might correspond to the “reading by hand” circuit (gesture recognition system) as opposed to the “reading by eye” circuit (shape recognition system in the VWFA). Furthermore, writing is not only learned in parallel with reading, but facilitates reading acquisition in children (Bara et al., 2004). Thus, the involvement of the left MFG in French children, although less important than in Chinese children, might be due to a heavier dependence on handwriting during reading acquisition than during adulthood. This interpretation is supported by a recent study showing that, in second-language learners of Chinese, viewing characters that were learned through character writing induced greater activation in LMFG compared with those learned without character writing (Cao and Perfetti, 2016).”

Please find the full reviews attached for your reference:Reviewer #1:1) Reasoning stems from a fallacy – namely that the lack of evidence for a difference is evidence for a lack of difference. It is not.

We totally agree with the Reviewer but here through multiple analyses, we tried to provide the readers with all the neural evidence we obtained when these 4 groups of children were exposed to words. Indeed, we did not find strong evidence supporting large differences between Chinese and French children. We have added a paragraph on the limitations of our study. At least, we have no evidence of a major difference between Chinese and alphabetic languages.

2) It is problematic that the authors interpret BOLD signal/ brain activity as somehow equivalent to a mechanism or computation, without any attempt to constrain or define that computation.

We have changed the word “mechanisms” to the more neutral “neural correlates”. This is a study of reading circuits at the whole-brain level. Its aims are to identify the correlates of reading skills in young children, and to probe the putative differences between those networks in two different writing systems. The purpose is not to attempt to define the precise computation underlying each region. However, we have added a more thorough discussion of what this computation could be (e.g. the role of the MFG in encoding writing gestures).

3) The study of dyslexia is important, but here it clouds the inference waters about the neural substrate of reading – it cannot separate what the neural "causes" vs the neural "effects" of dyslexia are.

Indeed, this criticism applies to all fMRI studies. Whether or not the reduced activations in reading-impaired relative to typical readers reflect the causes or the consequences of dyslexia, they highlight the regions involved in reading. Causal inference is not the only means of progressing in a scientific domain. With respect to adult reading, we have both causal evidence (if a brain infarct destroys the VWFA, the patient can no longer read (pure alexia) (Déjerine, 1891)) and correlational evidence (when illiterates are presented with words, a reduced activation is observed in the VWFA relative to literates, which is a consequence of having not learn to read (Dehaene, et al., 2010)). Both arguments point to the VWFA as a key region in the reading circuit. We used the same logic here, and we do not understand how “it clouds the inference water about the neural substrate of reading”. If in two different writing systems, dyslexia (or insufficient reading skills) is associated with similarly reduced activations relative to the typical readers, it is logic to infer that the corresponding sites are jointly involved in reading acquisition in both writing systems.

This point was already stated in the previous version of the manuscript: “Given that the

VWFA is also under-activated by written words in illiterates (Dehaene, et al., 2010; Hervais-Adelman, et al., 2019) and in typical children before they learn to read (Dehaene-Lambertz, et al., 2018), its under-activation in impaired readers may simply reflect the lack of reading practice – in other words, it might be a consequence rather than a cause of the reading deficit.” and “Again, because this site (pSTG) is also under-activated during spoken language processing in illiterates and in preliterate children compared to adult and children readers (Dehaene, et al., 2010) we cannot ascertain whether its anomalous activation is a cause or a consequence of the reading impairment. Indeed, it does not have to be one or the other, but could be both, as phonological awareness is known to be both a predictor of future reading impairments (Hulme, et al., 2012; Melby-Lervåg, et al., 2012), and a consequence of learning to read in an alphabetical language (Morais, et al., 1986).”

As this citation shows, we did not over-interpret the findings we discovered. We never tried to explain the group differences we found between typical and dyslexic readers as causes of dyslexia. Thus, we disagree that our study “clouds the inference waters about the neural substrate of reading”.

Finally, although it is important to understand what causes dyslexia, we believe that, for intervention purposes, it is equally important to understand what are its consequences, and whether they vary across languages. In this way, we progress in our ability to provide effective theoretical bases and guidance for the intervention of children with reading impairment. In the domain of developmental neurological impairments, the causes of the deficits may well turn out to be impossible to cure, but it may still be possible to intervene and help those children compensate for their deficits.

4) The experiment is designed to rely on interactions for scientific inference, yet appears only to be powered to detect main effects – though no power analysis is presented. The Bayes factors reported are weak evidence for the null, and worse are highly dependent on the prior. Using a more informed prior (for example, from the existing literature cited here, such as 9,10,32 and many others) will decrease evidence for the null dramatically (and killing this Bayes factor further, I suspect; moreover, it is only Bayes factors > 10 that are regarded as strong evidence for the null). That has not been done here. Ideally one needs to show a comparable range of Bayes factors with multiple, varied, or highly justified priors. That has not been done here. Mixed models should be used here over anova in order to take into account item variance simultaneously with subject variance.

Please see the response to the previous reviewer. Also, our analysis already uses one-sided alternative hypothesis as more informed prior, such as group1 (control) > group2 (impaired readers) and group 1 (Chinese) > group 2 (French) so that the Bayes factor we reported before were not being further killed. Besides, according to previous literature, evidence for H1 against H0 is generally considered as moderate for BF10≥3, and strong for BF10≥10. In the main analysis, the Bayes factors are larger than 9.9 in the FFG, MFG and STS. Post-hoc comparisons in each writing system show effects in the same direction. Given the literature, we were expecting large differences between Chinese and French in the VWFA and MFG except paradoxically the MFG in Chinese children. Following Editor’s suggestion, we have added a sensitivity analysis which confirmed that our Bayes factors are not largely driven by the priors we choose.

5) It is not clear how this work is an advance over existing findings, see in particular citations 9, 10, and 32.

Citation 9 (Rueckl, et al., 2015) and citation 10 (Szwed, et al., 2014) are cross-cultural fMRI studies based on adults who are proficient readers; impaired readers were not included in these two studies. Citation 32 (Goswami, et al., 2011) was a cross-cultural behavioral study that compared children with and without dyslexia in different writing systems but without fMRI data.

The reasons why we think our work is more advanced than previous studies are as follows:

a) First, the fact that the neural bases of different writing systems have already been compared in adults does not reduce the interest of performing similar (but more demanding) studies in children. Here, we study the mechanisms of learning which, in an immature and inexperienced brain, could conceptually be based on different mechanisms than those observed in the mature brain. This is especially the case for a highly cultural, education-dependent activity such as reading. In principle, brain activations in children could be much more distributed or, on the contrary, more restrained than in adults. Stability across ages, experience and culture is thus a remarkable fact. It is not such a trivial observation, if we compare this stability with the plasticity of the human brain, which allows for a considerable degree of reorganization after early brain injury (Cohen et al., 2004).

b) Second, it is still an ongoing debate about whether the neural network for reading is universal or different across languages. Although citation 9 and 10 consistently revealed the language-universal neural bases underlying proficient reading and citation 32 found that the common cognitive deficits in dyslexia across different writing systems, there are several studies that claim to provide support for the language-specific view of reading and reading disorders (Siok, et al., 2020; Siok, et al., 2004). Thus, the current literature is inconclusive about the question we asked in our work: to what extent are the brain mechanisms for reading acquisition similar across writing systems, and do the same neural abnormalities underlie developmental dyslexia in alphabetic and ideographic systems? However, up to now, there are very few comparable studies, probably owing to the difficulty of scanning large groups of children, and of organizing strictly comparable multi-site fMRI studies.

c) Last, we are in the best possible experimental situation to perform cross-linguistic comparisons: our groups are perfectly paired, very homogeneous in age and comprise a large number of children (24 in each group, for a total of 96 children; this is no small feat when considering the constraints that face such developmental studies). We used the same experimental task and brain imaging acquisition parameters in France and China, thus controlling for many confounding factors.

Reviewer #2:SummaryIn the current manuscript, authors show convergent results of common neural correlates of word recognition in typical developing and dyslexic children in French and Chinese. While their conclusion is based on "null effects" (i.e., Chinese and French dyslexics are similar), they use a set of exhaustive approaches (whole-brain MRI, ROIs, MVPA) and Bayesian inferences to support their findings. The results provide additional insight into the ongoing debate about the universality of the neural correlates of dyslexia across cultures and writing systems. Their results also shed light on the neural correlates of pediatric dyslexia in children showing similar characteristics to adults (though they do not compare with adults directly in their study). The study fills an important gap that goes beyond prior meta-analytic literature that showed that the neural basis of dyslexia is similar across alphabetic writing systems that are different across orthographic depths in the visual word form area (VWFA); they show using empirical data (and not meta-analyses) that even completely different writing systems such as Chinese and French are similar. The authors make pain-staking and impressive efforts to match across French and Chinese in terms of the various parameters (scanner characteristics and demographic). The results are generally convincing and manuscript well written. Detailed major comments are listed below.1) Premise of the study: The major issue for this reviewer was that the "universality of dyslexia" argument lies on the fact that there is some neural signature of dyslexia. Yet, at the whole brain level, there is no significant difference between dyslexia and controls. The rest of the analyses heavily relies on the regions identified in the ROI based approach. This should be thoroughly discussed and addressed as an important limitation.

We have added a paragraph on the limitations of our study at the end of the discussion. Nevertheless, we want to point out that testing the literature predictions by looking at specific ROIs is a valid way to improve statistical sensitivity by both decreasing the impact of corrections for repeated measures and by averaging activations on a set of voxels. Furthermore, we provided the figures with non-corrected activations to provide the reader with a complete information.

2) Correlation analyses with reading: Authors then move into correlation with reading ability but do not pursue further than showing correlations in the entire sample. They do not break the sample up into writing systems. Since authors do show significant correlation with reading abilities in several areas, there appears to be room to explore differences in writing systems. Can the authors do further analyses and potentially build on these results (also), and if not, justify why?

Following the suggestion, we did the correlation analysis in each writing system and added the results in the manuscript. As we can see from the Figures 2 B and C, similar regions (left STS, bilateral PCG, bilateral MFG, bilateral FFG) were sensitive to the reading performance in both languages.

3) Power & sample size: Discussion paragraph two: Authors mention that they were sufficiently powered because they had N=96. This is still 24 per group for each of the four groups. On what basis was it sufficiently powered? Especially in reporting these "null" results, it's important to provide evidence and perhaps reiterate this evidence when mentioning their "sufficient power" in the Discussion.

Please see response to the Editors’ point 4.

4) MVPA: In general MVPA was confusing in many ways and the reviewer recommends a major rewrite.a) Motivation: Introduction paragraph seven: The phrase about MVPA where they motivate this analysis should probably be a separate sentence as the question being asked is very different than the first part of the question. Further, this justification comes out of nowhere and should be elaborated. The authors claim that they wish to test dispersion and intensity using classical analyses, and also using MVPA. They also do not mention in the Introduction, but in the Materials and methods, mention that they are examining stability (reproducibility from Run 1 to Run 2). Please clarify in the Introduction.

We have rephrased and clarified why and how we did the MPVA analysis in the Introduction section.

“Because group analyses leave open the possibility that the observed group differences might be due to spatially more variable activations in impaired-readers than in typical readers, we also performed individual-based analyses and compared the location and activation values of the most responding voxels in each child. Finally, because classical analyses may mask the presence of fine-grained activity patterns that are specific to a given subject or a given category, we also quantified the stability of subject-specific activation patterns within reading-related ROIs using multivariate pattern analyses. The goal of these analyses is to circumvent the blurring effect of group analyses, which may hinder the discovery of genuine but more dispersed activations in reading impaired children relative to controls.”

b) Materials and methods subsection “Multivariate Pattern Analysis (MVPA”: This was very confusing all around. For example… (1) What values are fed into as features? Time-series correlation? Or betas for each trial? They mention "pattern of response evoked by words relative to fixation" but this is unclear. (2) They claim they are doing multi-"voxel" pattern analyses (MVPA), but they also seem to refer to each ROI having one correlation coefficient. (3) It was not clear why this is called MVPA. What was the analytical technique used, ANOVA? Correlation-based MVPA (what is this exactly?)? There are probably many more, but it's hard to judge the quality of this analyses based on their lack of details and hard to ask further questions at this point.

We first computed the correlation of the activations (beta values) of all voxels within a ROI of run1 with the activations of the same voxels in run 2, either within a given visual category (e.g. Word-Word) or between categories (e.g. [Word-Face, Word-House, Face-House]). Then an ANOVA was performed to compare the correlation coefficients within a category vs between categories. This was done for each ROI and separately for the Word category and the Face category. We have rephrased the Materials and methods and Results sections.

5) Population: The test scores in Chinese was normed based on their own data of 2554 individuals. It is not clear whether this is appropriate (though size is large). How were these children recruited; is this a nationally representative population? In addition, how French children with dyslexia was identified (and criteria) was not mentioned in this paper and the authors simply refer to another paper. Since this is quite important, it will be nice to have this detailed here in this manuscript as well.

The 2554 Chinese children were recruited from eight public primary schools in Beijing. Of these schools, five were located in the downtown area (in two districts whose total GDP ranked 1st and 2nd out of Beijing's 16 districts in 2017) and the last three in rural areas (in two districts whose total GDP ranked 8th and 13th out of Beijing's 16 districts in 2017). All Children in 3rd – 5th grades in each school were tested as a group in their classroom. Among the 2554 children, there were 368 3rd graders, 1158 4th graders and 1061 5th graders. Test results were standardized on these 2554 children because the previous standardized tests were based on 1993 data (e.g. Chinese Character Recognition Test) and were no more suitable for screening for dyslexia in 2013 because children can read many more characters than their counterparts of 20 years ago. Given that Beijing is one of the most developed cities in China, both economically and educationally (www.bjstats.gov.cn/tjsj/tjgb/ndgb/201511/t20151124_327764.html), it is difficult to determine whether these 2554 children could represent the national population, but we think they could at least represent Beijing.

Following the suggestion of Reviewer 3 and of the Editor, we have replaced "dyslexics" with "impaired readers" and "children with reading impairments" because the standard deviation we used (<-1.25) is not a strict criterion for dyslexia.

We have added the criteria for identifying French impaired readers in the Materials and methods section.

6) Reading ability: (1) How did they combine reading skills across writing systems to combine all in a correlation analysis, especially when reading measures that fed into the dyslexia criteria were different, there is no standardized measures in Chinese and just that the measures are so different across the board? (2) This also means that although unreported, the authors can compare reading scores (as well as others such as IQ) across writing systems, and report in Table 1? Please elaborate.

1) In the correlation analysis across all participants, we used standard scores in the main screening test of dyslexia in each language (Chinese character recognition test for Chinese children and “L’alouette” for French children). Although the tests are not equivalent, z-scoring allows to index each child reading lag relative to a normal cohort. The Chinese character recognition test (Wang and Tao, 1993) has been widely used to identify Chinese dyslexia (Feng, et al., 2017; Liu, et al., 2012; Qi, et al., 2016) and “L’alouette” is a standardized reading test classically used to detect dyslexia in French-speaking children (Altarelli, et al., 2014). These two tests are time-limited with penalties for errors.

2) Instead of comparing the reading scores, we added the CI 95%, which offers a better image of the values distribution in each group. The French reading impaired group can be considered as genuinely dyslexic (CI 95% [-2.34 -1.94] SD), whereas the Chinese group is less deviant (CI95% [ -1.54 -1.95]).

Other measures were not done systematically in all groups. For example, for French children, we considered children with normal reading scores who were in their appropriate academic level given their age with no complaint from the parents or the teacher, as typical children. A full IQ battery was performed in dyslexic French children but in a learning dedicated center at distance from the scan day. At the scan day, we only performed some subtests of Wechsler's WISC III or IV to ensure that they were in the normal range for all children (e.g. cubes for PIQ).

Chinese children were tested with the Raven's Progressive Matrices. The goal of these tests was only to dismiss any child with general cognitive difficulties. The z-score values of PIQ in these tests (with CI95%) are in Author response table 1:

**Author response table 1. resptable1:** 

	Mean	CI 95%
Chinese typical readers	0.67	0.44 ~ 0.89
Chinese impaired readers	0.69	0.47 ~ 0.91
French typical readers	-0.41	-0.85 ~ 0.36
French impaired readers	-0.11	-0.63 ~ 0.41

Reviewer #3:General Assessment. An interesting paper with rich results, an fMRI study comparing previously reported results of French children with new data on Chinse children. The study is characterized by careful sample matching, comparable instrumentation across sites, and multiple sophisticated analyses. The paper provides a context of previous brain imaging studies that compare dyslexia across Chinese and alphabetic writing and places its results in this context. Its message is not novel-that the brain shows a reading network that is shared across languages and writing systems-but because the research has suggested different conclusions on whether reading Chinese and reading alphabetic writing involve different brain areas, these results could add weight to two aspects of the issue. Do alphabetic and Chinese reading engage the same brain areas to the same extent? Do Chinese and alphabetic dyslexics show similar activation patterns across brain areas? The results answer the first question by "yes" and no", but the authors emphasize more the "yes" part. They answer the second part "Yes" Some revision and perhaps reframing of the contribution of the paper to the two issues engaged-language/writing system factors and dyslexia-would improve the paper.Major comments on context and Interpretation.1) One suggestion is to broaden the context for the question and to refine its form. Arguments that the procedures of reading are universal have been made on the basis of behavioral studies and the logic of writing systems (e.g. Perfetti, 2003; other observations by Mattingly, 1972) If graphs-the written symbols of language-encode linguistic forms-then the brain, to decode writing, must connect visual areas to language areas. Beyond that fundamental universal, the factors of mapping of phonology and morphology, and perhaps graphic complexity, could influence the details. Whether there is a shared reading circuit across languages has not been the main question. Rather, the question has been, given a shared or universal circuit, what differences might one expect based on the features of the writing system and the language?

We thank the reviewer for this comment and have since implemented this suggestion. We have revised the Abstract and the Discussion section to emphasize the language differences we found in the current study.

2) Although the authors' argument is strong on the universalist side, the data show convergence with the results of Siok, Tan et al. in the greater LFMG and SPL activation for Chinese. Puzzling, as the authors note, is the difference in pSTG, also higher for Chinese, contrary to other results. Given these language differences, what does the failure to find a language x dyslexia interaction mean? Both normal and dyslexics in Chinese show the Chinese pattern (more activation in MFG and SPL) and both controls and dyslexics in French show the French pattern (Less MFG, Less SPL, and, surprisingly, pSTG), as shown in Figure 4.

We agree with the comment. However, as also pointed by the reviewer, the differences between writing systems are due to modulation, not to a radical difference in circuits.

3) With these language differences, an appropriate interpretative framing would be to emphasize writing system/language differences while finding no dyslexia differences across languages. The paper reports and comments on the differences findings but does not highlight them. The Abstract and conclusion should be written to reflect these findings.

We thank the reviewer for this comment and have since implemented this suggestion. We have revised the title, Abstract, Discussion and conclusion section to emphasize the language differences we found in the current study.

4) The LFMG difference might reflect the role of writing experience in Chinese, which may make frontal brain areas near the motor cortex, especially Exner's area, more active in reading. The authors cite the comparison of French and Chinese by Nakamura et al., 2012), but the evidence from Cao & Perfetti, 2017, is also relevant because it reports that the overlap of writing and reading brain areas is greater in Chinese than English and also in learners of Chinese as L2 who had learned through character writing compared with learners who had learned through pinyin.

We thank the reviewer for this comment and have since implemented this suggestion. We have discussed more the function of L.MFG, emphasizing that the greater involvement of this region in Chinese readers might be related to their massive writing experience which reflects culture-modulation on reading circuits. We also added the mentioned paper.

5) More generally on the issue of context and interpretation, the authors should consider the relevance of Perfetti, Cao, and Booth (2013), which provides a more nuanced framing of the universal/specialization issue and summarizes results from children's studies carried out by Booth et al. to identify brain regions that show developmental changes in English and Chinese, highlighting commonalities along with divergence in STG and LMFG.

We thank the reviewer for this comment and we have reframed the manuscript.

6) Any comments on why MFG shows a much stronger dyslexia effect for French (strong) than Chinese children (marginal or null), according to the Bayes factor data shown in Table 4? The language difference shows more MFG involvement among Chinese than French. The reporting of the results in subsection “ROI-Level analysis” refers only to the Bayesian evidence against a language x dyslexia interaction, ignoring the post-hoc analyses (which given the goal of the study, could have been planned comparisons), which also show strong F10 support for an MFG difference between French and Chinese dyslexia but marginally "moderate" support for such a difference between French and Chinese controls.

We have added the Bayes factors of post-hoc analyses between controls and impaired readers in each language. A much stronger dyslexia effect for French (strong) than Chinese children (marginal or null) in LMFG may due to the fact the French impaired readers (<-2 SD) in the current study were more severe than Chinese impaired-readers in reading performance (<-1.25 SD).

7) Concerning the MFG, a fair summary of the results is that MFG activation was observed more for Chinese than French and more for controls than dyslexics. It would be appropriate to point out that this confirms conclusions from other research. Thus, the conclusions from Siok, Tan, et al. that the LMFG is more involved in reading Chinese than in alphabetic reading is also found here, as is their conclusion of its under-activation in Chinese dyslexia. What may be new in the present paper is evidence that under-activation of the MFG may also characterize reading French. This last conclusion is not based on the present results, however, but on results from the previously reported results for the French children. The authors should point out whether the original publication of the French results reached this conclusion or whether it emerged only in the comparisons of the present paper. The Monzalvo et al. abstract emphasizes VWFA factors in dyslexia, but not MFG.

We thank the reviewer for this comment and have since implemented this suggestion. We have rephrased and emphasized that the language effect and reading skill effect in LMFG in the current study is consistent with the conclusions from previous studies. We also mentioned the original publication of the French children in which we did observe different activation pattern of LMFG in controls and dyslexic children (see Monzalvo et al., 2012) although the difference was not significant at the whole brain level.

8) The Interpretation of the VWFA as a universal site should take into account that both faces and houses showed lower activation for dyslexics along the anterior-posterior axis. This affects how to interpret the lower activation for dyslexics to words in the VWFA.

The observation of a general reduction in activation in reading impaired children relative to controls, along the anterior-posterior axis does not modify the interpretation of the VWFA as a universal site of reading. We have added a short paragraph to discuss these findings:

“Thanks to individual peak location and intensity analyses, as well as multivariate pattern analyses, we could reject an alternative interpretation which, to the best of our knowledge, was not explicitly tested in previous studies: the possibility that the reduced activations are an artifact of group averaging, solely due to greater inter-individual variability in the localization of reading-related circuits in the dyslexic brain. Using individual peak, we observed that the brain localization to words were not more dispersed among dyslexic participants than among controls. Using MVPA, we showed that, within individual subjects, the activation patterns in the VWFA in response to written words were less reproducible across runs in dyslexics than in typical readers. This was solely the case for words, not for the other visual categories. We did observe a slightly reduced activation to faces and houses in dyslexics relative to controls, as previously reported in illiterate subjects (Dehaene et al., 2010), as well as seen in response to non-word stimuli (numbers, abstract strings) (Boros et al., 2016) and faces in dyslexics (Gabay et al., 2017; Monzalvo et al., 2012). However, the pattern of activity for faces was stable from one run to the next, contrary to what was found for words in the MVPA analyses. This observation suggests that the reduced activation to written words may not reflect a general disorganization of the extra-striate visual areas, but rather a specific difficulty with written words.”

9) The conclusion of the paper needs to be revised to more accurately reflect the results of the study. This should include the differences that were found, not just those that were not found and a more contextualized grounding for the conclusion that "cultural variability is merely reflected in the variable emphasis that different writing system put on (linguistic units)". This is not new (which is OK; it's worth emphasizing) and has been pointed in other papers in which the results have shown language differences.

We thank the reviewer for this comment that help us reframe the manuscript. We agree that we may have focused too much on the neural responses common to Chinese and French children without sufficient discussion of the differences. We have revised the Abstract and the Discussion section to emphasize the language differences we found in the current study.

10) The authors acknowledge a possible task issue but an additional consideration seems relevant. The task involved reading only incidentally. We know that for a skilled reader, implicit reading engages the network non-attentively (automatically), so the task itself is ok, as are naming and semantic judgement used in other studies. The problem is that, because it is the only task used, we get a limited picture of the reading network, perhaps especially for the dyslexics who cannot be assumed to engage the reading network automatically. It is worth noting that the research reporting Chinese-alphabetic differences did not use passive reading tasks but naming, rhyming, and meaning decision tasks. I don't know whether this is relevant for the differences, but It would strengthen the contribution of the paper if it could suggest a reason for different conclusions arising in the literature or address the task issue in that context.

We thank the reviewer for this comment and have since implemented this suggestion. In the revised manuscript, we pointed out in the Introduction and Discussion section that previous literature, particularly with Chinese dyslexics, used different kinds of reading tasks, e.g. picture and semantic matching task (Hu, et al., 2010), homophone judgment task (Siok, et al., 2004), font-size perceptual matching task (Siok, et al., 2009) and morphological task (Liu, et al., 2013), whereas our findings were based on a passive reading task. This may partly explain why we found a culturally-universal neurobiological basis of impaired reading, e.g. decreased activation in LMFG in both Chinese and French impaired readers while previous studies did not. We also discuss in greater detail the importance of the choice of a task where all children can perform equally well.

11) I also wonder about the degree of reading difficulties shown in the sample. The Chinese children were classified as dyslexic by showing a -1.25 SD score on the CCRT character recognition test. We need to know what the task was for the CCRT. Performance on a recognition task reflects learning through experience. A reading problem may reflect a poor learning outcome from experience but should not reflect mere experience. Also -1.25 SD seems to identify a rather large percentage of children as dyslexic, compared with other studies, which may make it more likely that reading experience is involved to some extent. This is not a problem for an approach that treats reading as a graded skill, which the authors do in some of their analyses. Making reading skill rather than dyslexia the framing of the paper could help with this.

We thank the reviewer for this comment and have since implemented this suggestion. Followed the Reviewer suggestion, taken up by the editor, we have made reading skill rather than dyslexia the framing of the paper. We used “impaired readers” and “reading-impaired children” rather than dyslexia to describe the group of poor readers and we focused on the effect of “reading skill” rather than “effect of dyslexia” in the manuscript as below -1.25 SD of the norm is not a strict criterion for dyslexia screening.

The Chinese character recognition test (CCRT) is a word-compounding test and consists of 210 single Chinese characters, which were divided into ten groups based on their difficulty. Based on each of the given characters/morphemes, the participants were asked to write down a compound word or phrase on an answer sheet, which included the given character or morpheme as a part. For example, they may write down the word of “池 塘 /chi2tang2/(meaning pond)” for the character of 池/chi2/(meaning pool). The participants were asked to writing down as many as possible in 45 min. The number of correct answers was the raw score. 2554 children from grades 3–5 were tested with this test for their own grades. The mean and standard deviation of the performance of character recognition were calculated for each grade.

12) The ANOVA reported in subsection “Data-driven Analyses” does not provide a main effect of category. Instead it tests one category against the mean of the other two. And this leads to multiple tests that are appropriate with a significant main effect. A standard ANOVA with 3 levels of the category variable is needed.

We were interested by category-specific activations contrasts (category X > mean of the other two categories) not the main effect of category. We corrected the text.

13) Another statistical issue is the use of separate planned comparisons within each language when no interaction involving language is reported. In subsection “Anterior-to-Posterior gradient in the visual cortex”, this comparison, without a report of a significant language interaction, is shown for the anterior-posterior gradient analysis allowing the suggestion that the only dyslexia difference for both languages was near the VFWA. In other analyses, the authors report a lack of a language interaction and no further within-language comparisons. Given the focus of the research on language differences, planned comparisons between languages is quite reasonable to do in all analysis but the authors seem to have done this only selectively. Some explanation is needed.

We have added the post-hoc within-language comparisons between control and impaired readers, as well as the between language comparisons when language× Reading skills interaction was not significant.

14) The sentences "Dyslexics might have more dispersed activations without focal peaks that would also create weaker responses at the group-level, but a reproducible pattern of activations. In that case, the within-subject reproducibility of multivariate activation patterns should not differ between normal-readers and dyslexics" from subsection “MVPA of the activations to words”. The logic here escapes me. If Ds have more dispersed activation without focal peaks their reproducibility of multivariate patterns could still differ from controls. More voxels but not the same voxels, reflecting intrinsic variability. I think the authors are right that the individual peaks analysis means the between-group variability is not due to averaging individuals. But the differences in these individual MVPA analyses mean that variability is higher with dyslexics. The reason for this should be considered.

We have entirely rewritten this paragraph to clarify the logic in both Introduction and Results sections.

“Because group analyses leave open the possibility that the observed group differences might be due to spatially more variable activations in impaired-readers than in typical readers, we also performed individual-based analyses and compared the location and activation values of the most responding voxels in each child. Finally, because classical analyses may mask the presence of fine-grained activity patterns that are specific to a given subject or a given category, we also quantified the stability of subject-specific activation patterns within reading-related ROIs using multivariate pattern analyses. The goal of these analyses is to circumvent the blurring effect of group analyses, which may hinder the discovery of genuine but more dispersed activations in reading impaired children relative to controls.”

“The above analyses were carried out in a standardized way at the group level. It is therefore possible that the observed group differences were due to a greater inter-individual variability in brain localization in the reading-impaired group than in the control group. This possibility would lead to a completely different interpretation of the results: each child might have a well-organized brain activity for reading, with the only anomaly being a greater anatomical dispersion in the reading-impaired group compared to the control group. To test this possibility, we performed two individual-based analyses, one based on the comparison of the location and activation values of the most responding voxels and the other examining the stability of the pattern of responses across runs through a multi-voxel pattern analysis (MVPA). We focused on the ROIs previously showing significant differences due to Reading skill (i.e. left FFG, MFG, PCG and STS) and Language (i.e. left MFG, SPL and pSTG).”

15) Why on the MVPA are the data analyzed transformations of correlations of first run with second run? The idea is reproducibility, but the second run is a second run that provides a familiarization component that may affect controls and dyslexics differently.

The visual stimuli (except checkerboard) were different in each run. Furthermore, a familiarization component (i.e. due to the task itself) should similarly affect activation to words and faces.

16) Another point to be explained is why some analyses use fixation as the comparison (MVPA) whereas most use the contrast of one condition vs the other two.

When we were interested in category-specific/preferential activation, we used the contrast of (one category vs others). For example, in Figure 1A, we showed word-specific and facespecific activation maps as these regions had greater activation than the mean of other categories. When we were comparing children’s response to a visual category of interest, e.g. word response in the current study, we used the contrast of (words vs fixation). It allows to more clearly see the direction of the differences between children because for example a larger response to words in controls or a weaker response to others in dyslexics would induce a significantly larger effect in the contrast (word > others) * (controls > dyslexics).

17) Figure 2 shows areas with correlations with reading scores. Either the caption or the images should make clear the directions of the correlations and make clear this is for all subjects. Given the goal of the study, showing these separately for French and Chinese is better, even though it reduces the power.

We thank the reviewer for this comment and have since implemented this suggestion. We did the correlation analysis in each writing system and accordingly revised the figure.

As predicted by Reviewer 3, the power is reduced when data are split according to the language. To provide the readers with full information, we presented the results at the threshold of voxel-level p<0.005 and cluster-level uncorrected. Figure 2B and 2C separately displayed regions positively correlated with reading scores in Chinese and French children. As we can see from the figures, similar regions (left STS, bilateral PCG, bilateral MFG, bilateral FFG) were sensitive to the reading performance in the two languages.

18) Figure 3A is a bit puzzling. It shows peaks from literature on Chinese and alphabetic studies, but the only peak in LH is at MFG for Chinese. Certainly, studies of Chinese find peaks in other LH regions. Do the peaks have the same definition across studies and meta-analyses? One peak for each region of interest or region of a hypothesized reading network probably would show a different picture. As it stands the visual here contradicts universality and may be misleading. As for 3B, again it would be useful to indicate the direction of the correlations or state they are all positive if that is the case.

a) In ROI analysis, we aimed to test regions who had been reported sensitive to dyslexia/impaired reading in previous literature with our data. We thus first searched metaanalyses of imaging studies in dyslexics rather than in typically developing readers. However, due to the limited number of published neuroimaging studies of Chinese dyslexia, no metaanalysis was available to summarize the available evidence of into a pooled estimate. However, atypical activation in a lateral prefrontal region within BA 9 has been reported in readingimpaired Chinese children (Siok et al., 2004; Siok et al., 2009) and this region was repeatedly found to be more involved in reading Chinese than alphabetic languages (Bolger et al., 2005; Nakamura et al., 2012; Tan et al., 2005a). Besides, previous studies also often reported that Chinese reading networks are more symmetrical in the ventral visual system. We thus included the foci in both left middle frontal gyrus and right occipital cortex that were reported in several meta-analyses on Chinese typical reading (Bolger et al., 2005; Tan et al., 2005a; Wu et al., 2012; Zhu et al., 2014) and created representative ROIs. Notably, all ROIs (except two ROIs in the right-hemisphere) fell within the reading circuit identified in our participants (Words > other categories; see Figure 4 and Figure 4—figure supplement 1).

b) We have revised the Figure 3. We used red and blue dots to separately represent ROIs whose activation to words versus fixation were positively and negatively correlated with reading scores across all participants. Gray dots represent ROIs that did not reach significance in the correlation analysis.

19) The authors say that the LFMG dysfunction "was previously claimed to be specific to Chinese dyslexia". This is more accurately described as a discovery of an LMFG deficit in Chinese dyslexics in the absence of any existing evidence of a comparable deficit having been reported in in studies of alphabetic languages. The previously published study of French children should have added the discovery that LFMG dysfunction can characterize alphabetic reading as well.

We thank the reviewer for this comment and have rephrased in the revised manuscript.

Besides, we did observe different activation pattern of LMFG between controls and dyslexic children in our original publication of the French children (see Monzalvo et al., 2012) although the difference was not significant at the whole brain level. We mentioned this in the Discussion section.

[Editors' note: further revisions were suggested prior to acceptance, as described below.]

Revisions for this paper:Please address points 1-4 by the reviewer; each query is concrete, concise, addressable, and clearly laid out. Please make these revisions and clearly indicated in the revised manuscript text as well as describe in a response letter how you have handled each point.Reviewer #3:The authors made several significant improvements in the paper and were responsive to reviewers' comments. I think the strength of the paper lies in the developmental results for a Chinese sample to compare with previously published results of a French sample, along with a better portrayal of its contribution in the context of other research. The French-Chinese comparison itself remains a question for me, but the Abstract, Discussion, and conclusion are more in line with the results.I suggest the authors consider a few additional revisions.1) The authors' reframing from dyslexia to reading skill is important, given the likely differences in reading skill between the Chinese and French samples. But the use of the word "impaired" to refer to lower skill readers throughout the ms does not follow through on this reframing. Dyslexia is an impairment. It might be that the French sample includes mainly or exclusively impaired readers and the authors may prefer that term to be consistent with their original publication of the French data. If so, it might be enough to have a sentence or two that explicitly address the terminology shifts or point out the possible French-Chinese differences in the low skill group. If effective reading experience is what is involved especially in Chinese-i.e. the rate at which a child has benefitted from learning opportunities, including reading practice-this might be important to mention. This would be consistent with the demands of incremental character learning and practice. It may be interesting that the distinction between impairment and delay may not matter given the authors' interpretation of the results of their comparisons.

We agree with the Reviewer’s suggestion and replaced the term “impaired readers” by “poor readers”. In the participants section, we re-introduced the term dyslexics to describe the French children, as these children met the diagnostic criteria for dyslexia, but we pointed out, as suggested, that the Chinese poor readers might not be proper dyslexics as French children were.

We replaced the word “controls” by “typical readers” in the figures to be consistent with the text.

2) If I understand correctly, the important conclusion that LMFG is part of the universal network is indirect, dependent on adding a Chinese sample to the original French sample for comparison. The French result did not show significant under-activation for impaired readers. Instead, only when the Chinese sample was included did this occur. I think this point should be made explicit. Maybe I misunderstand something, but it seems that only by adding a Chinese sample that we see LMFG problem in French children with reading problems.

We thank the reviewer for this comment and apologize that we were not clear but his/her statement is not correct.

First, we did observe different activation patterns of LMFG in dyslexics and controls in our original publication of the French children (please refer to Figure 1 and Table 3 in Monzalvo et al., 2012), but no difference survived correction for repeated measures at the whole brain level. Because we were interested at that time by the organization of the visual ventral regions, we restricted our analyses to a ventral mask for the comparisons between dyslexic and typical readers. Thus, we did not emphasize the putative LMFG difference in this paper. The other paper, Altarelli et al. 2013, was also only focused on the occipito-temporal region. Here in the new set of French children (The 24 children were chosen in both previous sets in function of the Chinese children age and gender), we observed an effect of reading performance in the LMFG (see Figure 2). When the French participants are split in typical and dyslexic readers, we obtained a significant cluster (33 voxels, pFWE_corr = 0.007, Z = 4.46 at [-54 12 30]) showing larger activation in typical readers than in dyslexics.

Second, in the ROI analysis in the current study, both the frequentist and Bayesian analyses revealed that the reading skill effect in LMFG was robust in French children.

Therefore, the LMFG is a genuine site also in the French participants which has been underestimated due to previous analyses strategies. It cannot be claimed that this site is specific to Chinese reading.

3) The greater activation for Chinese than French in the posterior superior temporal gyrus/sulcus continues to be a bit of puzzle. The explanation on 19-20 does not seem very plausible. Children's use of PinYin might help explain why the PSTG in Chinese is not less activated than in French, contrary to expectations based on decoding accounts of the PSTG. However, I don't see how it can explain the reversal that was observed for children who have been reading characters for 3-5 years. Rather than this extended speculation, it might be better to just mention it as one of several possibilities, including the possibility of more complex syllable decoding because of Chinese tones and connections to spoken language functions, or a need to suppress the activation of the syllable associated with the character (because it leads to competition for character identification), or others that also are speculative. If the evidence for a letter-phoneme function of this area comes mainly from explicit reading tasks, maybe this area is doing something else related to the target task during incidental reading.

We thank the reviewer for this comment and have added sentences following his/her suggestions.

4) The authors use "script" where they actually refer to writing system. I strongly suggest a change in the title and elsewhere to writing system. The difference between French and Chinese is one of writing system and this inevitably brings along a difference in script. The difference between French Cursive and French Gothic and Chinese Traditional and Chinese Simplified are differences of script.

We have reviewed the entire manuscript to have a consistent terminology. The revised title is now “A universal reading network and its modulation by writing system and reading ability in French and Chinese children.”

**References**

Altarelli, I., Leroy, F., Monzalvo, K., Fluss, J., Billard, C., Dehaene-Lambertz, G., Galaburda, A.M., Ramus, F. (2014) Planum temporale asymmetry in developmental dyslexia: Revisiting an old question. Hum Brain Mapp, 35:5717-35.

Cherodath, S., Singh, N.C. (2015) The influence of orthographic depth on reading networks in simultaneous biliterate children. Brain & Language, 143:42-51.

Déjerine, J. (1891) Sur un cas de cécité verbale avec agraphie suivi d’autopsie. Mémoires de la Société de Biologie, 3:197-201.

Dehaene-Lambertz, G., Monzalvo, K., Dehaene, S. (2018) The emergence of the visual word form: Longitudinal evolution of category-specific ventral visual areas during reading acquisition. PLoS biology, 16:e2004103.

Feng, X., Li, L., Zhang, M., Yang, X., Tian, M., Xie, W., Lu, Y., Liu, L., Belanger, N.N., Meng, X., Ding, G. (2017) Dyslexic Children Show Atypical Cerebellar Activation and Cerebro-Cerebellar Functional Connectivity in Orthographic and Phonological Processing. Cerebellum, 16:496-507.

Liu, L., Wang, W., You, W., Li, Y., Awati, N., Zhao, X., Booth, J.R., Peng, D. (2012) Similar alterations in brain function for phonological and semantic processing to visual characters in Chinese dyslexia. Neuropsychologia, 50:2224-32.

Mumford, J.A. (2012) A power calculation guide for fMRI studies. Social cognitive and affective neuroscience, 7:738-742.

Pinel, P., Lalanne, C., Bourgeron, T., Fauchereau, F., Poupon, C., Artiges, E., Le Bihan, D., Dehaene-Lambertz, G., Dehaene, S. (2015) Genetic and Environmental Influences on the Visual Word Form and Fusiform Face Areas. Cerebral Cortex, 25:2478-2493.

Pleisch, G., Karipidis, I.I., Brauchli, C., Röthlisberger, M., Hofstetter, C., Stämpfli, P., Walitza, S., Brem, S. (2019) Emerging neural specialization of the ventral occipitotemporal cortex to characters through phonological association learning in preschool children. NeuroImage, 189:813-831.

Qi, T., Gu, B., Ding, G.S., Gong, G.L., Lu, C.M., Peng, D.L., Malins, J.G., Liu, L. (2016) More bilateral, more anterior: Alterations of brain organization in the large-scale structural network in Chinese dyslexia. Neuroimage, 124:63-74.

Siok, W.T., Jia, F., Liu, C.Y., Perfetti, C.A., Tan, L.H. (2020) A Lifespan fMRI Study of Neurodevelopment Associated with Reading Chinese. Cerebral Cortex.